# Gradient Starvation:
# A Learning Proclivity in Neural Networks

**Mohammad Pezeshki**[1,2]    **Sékou-Oumar Kaba**[1,3]    **Yoshua Bengio**[1,2]
**Aaron Courville**[1,2]    **Doina Precup**[1,3,4]    **Guillaume Lajoie**[1,2]

[1]Mila    [2]Université de Montréal    [3]McGill University    [4]Google DeepMind

corresponding authors:{pezeshki, guillaume.lajoie}@mila.quebec

## Abstract

We identify and formalize a fundamental gradient descent phenomenon leading to a learning proclivity in over-parameterized neural networks. *Gradient Starvation* arises when cross-entropy loss is minimized by capturing only a subset of features relevant for the task, despite the presence of other predictive features that fail to be discovered. This work provides a theoretical explanation for the emergence of such feature imbalances in neural networks. Using tools from Dynamical Systems theory, we identify simple properties of learning dynamics during gradient descent that lead to this imbalance, and prove that such a situation can be expected given certain statistical structure in training data. Based on our proposed formalism, we develop guarantees for a novel but simple regularization method aimed at decoupling feature learning dynamics, improving accuracy and robustness in cases hindered by gradient starvation. We illustrate our findings with simple and real-world out-of-distribution (OOD) generalization experiments.

## 1 Introduction

In 1904, a horse named *Hans* attracted worldwide attention due to the belief that it was capable of doing arithmetic calculations [81]. Its trainer would ask Hans a question, and Hans would reply by tapping on the ground with its hoof. However, it was later revealed that the horse was only noticing subtle but distinctive signals in its trainer's unconscious behavior, unbeknown to him, and not actually performing arithmetic. An analogous phenomenon has been noticed when training neural networks [e.g. 85, 109, 54, 39, 17, 14, 37, 51, 107, 76, 48, 19, 61, 77]. In many cases, state-of-the-art neural networks appear to focus on low-level **superficial correlations**, rather than more abstract and robustly informative features of interest [16, 88, 40, 68, 30].

The rationale behind this phenomenon is well known by practitioners: given strongly-correlated and fast-to-learn features in training data, gradient descent is biased towards learning them first. However, the precise conditions leading to such learning dynamics, and how one might intervene to control this *feature imbalance* are not entirely understood. Recent work aims at identifying the reasons behind this phenomenon [97, 70, 22, 73, 51, 76, 100, 92, 83, 105, 42, 79, 4], while complementary work quantifies resulting shortcomings, including poor generalization to out-of-distribution (OOD) test data, reliance upon spurious correlations, and lack of robustness [30, 68, 77, 41, 63, 64, 9]. However most established work focuses on squared-error loss and its particularities, where results do not readily generalize to other objective forms. This is especially problematic since for several classification applications, cross-entropy is the loss function of choice, yielding very distinct learning dynamics. In this paper, we argue that *Gradient Starvation*, first coined in [26], is a leading cause for this *feature imbalance* in neural networks trained with cross-entropy, and propose a simple approach to mitigate it.

35th Conference on Neural Information Processing Systems (NeurIPS 2021).

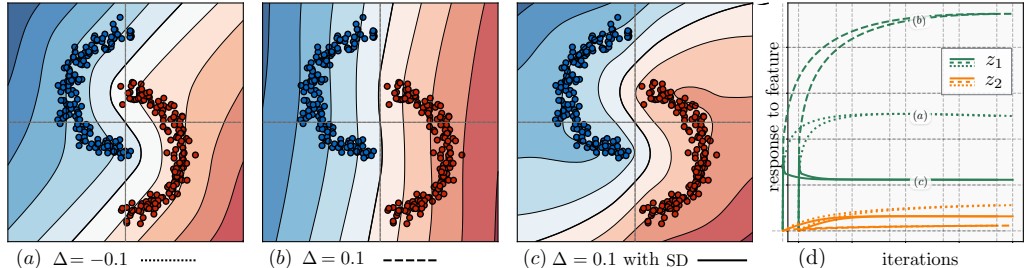

$(a)\ \Delta = -0.1$ ············  $(b)\ \Delta = 0.1$ - - - - -  $(c)\ \Delta = 0.1$ with SD ———  (d)  iterations

Figure 1: Diagram illustrating the effect of gradient starvation in a simple 2-D classification task. **(a)** Data is not linearly separable and the learned decision boundary is curved. **(b)** Data is linearly separable by a small margin ($\Delta = 0.1$). This small margin allows the network to discriminate confidently only along the horizontal axis and ignore the vertical axis. **(c)** Data is linearly separable as in (b). However, with the proposed Spectral decoupling (SD), a curved decision boundary with a large margin is learned. **(d)** Diagram shows the evolution of two of the features (Eq. 4) of the dynamics in three cases shown as dotted, dashed and solid lines. **Analysis:** (dotted) vs (dashed): Linear separability of the data results in an increase in $z_1$ and a decrease (starvation) of $z_2$. (dashed) vs (solid): SD suppresses $z_1$ and hence allows $z_2$ to grow. Decision boundaries are averaged over ten runs. More experiments with common regularization methods are provided in App. B.

Here we summarize our contributions:

- We provide a theoretical framework to study the learning dynamics of linearized neural networks trained with cross-entropy loss in a dual space.
- Using perturbation analysis, we formalize Gradient Starvation (GS) in view of the coupling between the dynamics of orthogonal directions in the feature space (Thm. 2).
- We leverage our theory to introduce Spectral Decoupling (SD) (Eq. 17) and prove this simple regularizer helps to decouple learning dynamics, mitigating GS.
- We support our findings with extensive empirical results on a variety of classification and adversarial attack tasks. All code and experiment details available at `GitHub repository`.

In the rest of the paper, we first present a simple example to outline the consequences of GS. We then present our theoretical results before outlining a number of numerical experiments. We close with a review of related work followed by a discussion.

## 2   Gradient Starvation: A simple example

Consider a 2-D classification task with a training set consisting of two classes, as shown in Figure 1. A two-layer ReLU network with 500 hidden units is trained with cross-entropy loss for two different arrangements of the training points. The difference between the two arrangements is that, in one setting, the data is not linearly separable, but a slight shift makes it linearly separable in the other setting. This small shift allows the network to achieve a negligible loss by only learning to discriminate along the horizontal axis, ignoring the other. This contrasts with the other case, where both features contribute to the learned classification boundary, which arguably matches the data structure better. We observe that training longer or using different regularizers, including weight decay [58], dropout [95], batch normalization [49], as well as changing the optimization algorithm to Adam [56] or changing the network architecture or the coordinate system, do not encourage the network to learn a curved decision boundary. (See App. B for more details.)

We argue that this occurs because cross-entropy loss leads to gradients "starved" of information from vertical features. Simply put, when one feature is learned faster than the others, the gradient contribution of examples containing that feature is diminished (i.e., they are correctly processed based on that feature alone). This results in a lack of sufficient gradient signal, and hence prevents any remaining features from being learned. This simple mechanism has potential consequences, which we outline below.

### 2.1   Consequences of Gradient Starvation

**Lack of robustness.**    In the example above, even in the right plot, the training loss is nearly zero, and the network is very confident in its predictions. However, the decision boundary is located very

close to the data points. This could lead to adversarial vulnerability as well as lack of robustness when generalizing to out-of-distribution data.

**Excessive invariance.** GS could also result in neural networks that are invariant to task-relevant changes in the input. In the example above, it is possible to obtain a data point with low probability under the data distribution, but that would still be classified with high confidence.

**Implicit regularization.** One might argue that according to Occam's razor, a simpler decision boundary should generalize better. In fact, if both training and test sets share the same dominant feature (in this example, the feature along the horizontal axis), GS naturally prevents the learning of less dominant features that could otherwise result in overfitting. Therefore, depending on our assumptions on the training and test distributions, GS could also act as an implicit regularizer. We provide further discussion on the *implicit regularization* aspect of GS in Section 5.

## 3 Theoretical Results

In this section, we study the learning dynamics of neural networks trained with cross-entropy loss. Particularly, we seek to decompose the learning dynamics along orthogonal directions in the feature space of neural networks, to provide a formal definition of GS, and to derive a simple regularization method to mitigate it. For analytical tractability, we make three key assumptions: (1) we study deep networks in the Neural Tangent Kernel (NTK) regime, (2) we treat a binary classification task, (3) we decompose the interaction between two features. In Section 4, we demonstrate our results hold beyond these simplifying assumptions, for a wide range of practical settings. All derivation details can be found in SM C.

### 3.1 Problem Setup and Gradient Starvation Definition

Let $\mathcal{D} = \{\mathbf{X}, \mathbf{y}\}$ denote a training set containing $n$ datapoints with $d$ dimensions, where, $\mathbf{X} = [\mathbf{x}_1, ..., \mathbf{x}_n] \in \mathbb{R}^{n \times d}$ and their corresponding class label $\mathbf{y} \in \{-1, +1\}^n$. Also let $\hat{\mathbf{y}}(\mathbf{X}) := f^{(L)}(\mathbf{X}) : \mathbb{R}^{n \times d} \to \mathbb{R}^n$ represent the logits of an L-layer fully-connected neural network where each hidden layer $h^{(l)}(x) \in \mathbb{R}^{d_l}$ is defined as follows,

$$\begin{cases} f^{(l)}(\mathbf{x}_i) = \mathbf{W}^{(l)} h^{(l-1)}(\mathbf{x}_i) \\ h^{(l)}(\mathbf{x}_i) = \sqrt{\frac{\gamma}{d_l}} \xi(f^{(l)}(\mathbf{x}_i)) \end{cases}, \, l \in \{0, 1, ..., L\}, \tag{1}$$

in which $\mathbf{W}^{(l)} \in \mathbb{R}^{d_l \times d_{l-1}}$ is a weight matrix drawn from $\mathcal{N}(0, \mathbf{I})$ and $\gamma$ is a scaling factor to ensure that norm of each $h^{(l-1)}$ is preserved at initialization (See [28] for a formal treatment). The function $\xi(.)$ is also an element-wise non-linear activation function.

Let $\boldsymbol{\theta} = \text{concat}\left(\cup_{l=1}^L \text{vec}(\mathbf{W}^{(l)})\right) \in \mathbb{R}^m$ be the concatenation of all vectorized weight matrices with $m$ as the total number of parameters. In the NTK regime [52], in the limit of infinite width, the output of the neural network can be approximated as a linear function of its parameters governed by the neural tangent random feature (NTRF) matrix [23],

$$\boldsymbol{\Phi}(\mathbf{X}, \boldsymbol{\theta}) = \frac{\partial \hat{\mathbf{y}}(\mathbf{X}, \boldsymbol{\theta})}{\partial \boldsymbol{\theta}} \in \mathbb{R}^{n \times m}. \tag{2}$$

In the wide-width regime, the NTRF changes very little during training [62], and the output of the neural network can be approximated by a first order Taylor expansion around the initialization parameters $\boldsymbol{\theta}_0$. Setting $\boldsymbol{\Phi}_0 \equiv \boldsymbol{\Phi}(\mathbf{X}, \boldsymbol{\theta}_0)$ and then, without loss of generality, centering parameters and the output coordinates to their value at the initialization ($\boldsymbol{\theta_0}$ and $\hat{\mathbf{y}}_0$), we get

$$\hat{\mathbf{y}}(\mathbf{X}, \boldsymbol{\theta}) = \boldsymbol{\Phi}_0 \boldsymbol{\theta}. \tag{3}$$

Dominant directions in the feature space as well as the parameter space are given by principal components of the NTRF matrix $\boldsymbol{\Phi}_0$, which are the same as those of the NTK Gram matrix [106]. We therefore introduce the following definition.

**Definition 1** (Features and Responses)**.** *Consider the singular value decomposition (SVD) of the matrix $\mathbf{Y}\boldsymbol{\Phi}_0 = \mathbf{U}\mathbf{S}\mathbf{V}^T$, where $\mathbf{Y} = diag(\mathbf{y})$. The $j$th **feature** is given by $(\mathbf{V}^T)_{j.}$. The **strength** of $j$th feature is represented by $s_j = (\mathbf{S})_{jj}$. Also, $(\mathbf{U})_{.j}$ contains the weights of this feature in all examples. A neural network's **response to a feature** $j$ is given by $z_j$ where,*

$$\mathbf{z} := \mathbf{U}^T \mathbf{Y} \hat{\mathbf{y}} = \mathbf{S}\mathbf{V}^T \boldsymbol{\theta}. \tag{4}$$

In Eq. 4, the response to feature $j$ is the sum of the responses to every example in $(\mathbf{Y}\hat{\mathbf{y}})$ multiplied by the weight of the feature in that example $(\mathbf{U}^T)$. For example, if all elements of $(\mathbf{U})_{.j}$ are positive, it indicates a perfect correlation between this feature and class labels. We are now equipped to formally define GS.

**Definition 2** (Gradient Starvation). *Recall the the model prescribed by Eq. 3. Let $z_j^*$ denote the model's response to feature $j$ at training optimum $\boldsymbol{\theta}^*$[1]. Feature $i$ **starves the gradient** for feature $j$ if $dz_j^*/d(s_i^2) < 0$.*

This definition of GS implies that an increase in the strength of feature $i$ has a detrimental effect on the learning of feature $j$. We now derive conditions for which learning dynamics of system 3 suffer from GS.

### 3.2 Training Dynamics

We consider the widely used ridge-regularized cross-entropy loss function,

$$\mathcal{L}(\boldsymbol{\theta}) = \mathbf{1} \cdot \log\left[1 + \exp\left(-\mathbf{Y}\hat{\mathbf{y}}\right)\right] + \frac{\lambda}{2}\|\boldsymbol{\theta}\|^2, \tag{5}$$

where $\mathbf{1}$ is a vector of size $n$ with all its elements equal to 1. This vector form simply represents a summation over all the elements of the vector it is multiplied to. $\lambda \in [0, \infty)$ denotes the weight decay coefficient.

Direct minimization of this loss function using the gradient descent obeys coupled dynamics and is difficult to treat directly [26]. To overcome this problem, we call on a variational approach that leverages the Legendre transformation of the loss function. This allows tractable dynamics that can directly incorporate rates of learning in different feature directions. Following [50], we note the following inequality,

$$\log\left[1 + \exp\left(-\mathbf{Y}\hat{\mathbf{y}}\right)\right] \geq H(\boldsymbol{\alpha}) - \boldsymbol{\alpha} \odot \mathbf{Y}\hat{\mathbf{y}}, \tag{6}$$

where $H(\boldsymbol{\alpha}) = -\left[\boldsymbol{\alpha}\log\boldsymbol{\alpha} + (1-\boldsymbol{\alpha})\log(1-\boldsymbol{\alpha})\right]$ is Shannon's binary entropy function, $\boldsymbol{\alpha} \in (0,1)^n$ is a variational parameter defined for each training example, and $\odot$ denotes the element-wise vector product. Crucially, the equality holds when the maximum of r.h.s. w.r.t $\boldsymbol{\alpha}$ is achieved at $\boldsymbol{\alpha}^* = \frac{\partial\mathcal{L}}{\partial(\mathbf{Y}\hat{\mathbf{y}})^T}$, which leads to the following optimization problem,

$$\min_{\boldsymbol{\theta}} \mathcal{L}(\theta) = \min_{\boldsymbol{\theta}} \max_{\boldsymbol{\alpha}} \left(\mathbf{1} \cdot H(\boldsymbol{\alpha}) - \boldsymbol{\alpha}\mathbf{Y}\hat{\mathbf{y}} + \frac{\lambda}{2}\|\boldsymbol{\theta}\|^2\right), \tag{7}$$

where the order of min and max can be swapped (see Lemma 3 of [50]). Since the neural network's output is approximated by a linear function of $\boldsymbol{\theta}$, the minimization can be performed analytically with an critical value $\boldsymbol{\theta}^{*T} = \frac{1}{\lambda}\boldsymbol{\alpha}\mathbf{Y}\boldsymbol{\Phi}_0$, given by a weighted sum of the training examples. This results in the following maximization problem on the dual variable, i.e., $\min_{\theta}\mathcal{L}(\theta)$ is equivalent to,

$$\min_{\boldsymbol{\theta}} \mathcal{L}(\boldsymbol{\theta}) = \max_{\boldsymbol{\alpha}} \left(\mathbf{1} \cdot H(\boldsymbol{\alpha}) - \frac{1}{2\lambda}\boldsymbol{\alpha}\mathbf{Y}\boldsymbol{\Phi}_0\boldsymbol{\Phi}_0^T\mathbf{Y}^T\boldsymbol{\alpha}^T\right). \tag{8}$$

By applying continuous-time gradient ascent on this optimization problem, we derive an autonomous differential equation for the evolution of $\boldsymbol{\alpha}$, which can be written in terms of features (see Definition 1),

$$\dot{\boldsymbol{\alpha}} = \eta\left(-\log\boldsymbol{\alpha} + \log(1-\boldsymbol{\alpha}) - \frac{1}{\lambda}\boldsymbol{\alpha}\mathbf{U}\mathbf{S}^2\mathbf{U}^T\right), \tag{9}$$

where $\eta$ is the learning rate (see SM C.1 for more details). For this dynamical system, we see that the logarithm term acts as barriers that keep $\alpha_i \in (0,1)$. The other term depends on the matrix $\mathbf{U}\mathbf{S}^2\mathbf{U}^T$, which is positive definite, and thus pushes the system towards the origin and therefore drives learning.

When $\lambda \ll s_k^2$, where $k$ is an index over the singular values, the linear term dominates Eq. 9, and the fixed point is drawn closer towards the origin. Approximating dynamics with a first order Taylor expansion around the origin of the second term in Eq. 9, we get

$$\dot{\boldsymbol{\alpha}} \approx \eta\left(-\log\boldsymbol{\alpha} - \frac{1}{\lambda}\boldsymbol{\alpha}\mathbf{U}\left(\mathbf{S}^2 + \lambda\mathbf{I}\right)\mathbf{U}^T\right), \tag{10}$$

with stability given by the following theorem with proof in SM C.

---

[1]Training optimum refers to the solution to $\nabla_{\boldsymbol{\theta}}\mathcal{L}(\boldsymbol{\theta}) = 0$.

**Theorem 1.** *Any fixed points of the system in Eq. 10 is attractive in the domain $\alpha_i \in (0, 1)$.*

At the fixed point $\boldsymbol{\alpha}^*$, corresponding to the optimum of Eq. 8, the feature response of the neural network is given by,

$$\mathbf{z}^* = \frac{1}{\lambda}\mathbf{S}^2\mathbf{U}^T\boldsymbol{\alpha}^{*T}. \tag{11}$$

See App. A for further discussions on the distinction between "feature space" and "parameter space". Below, we study how the strength of one feature could impact the response of the network to another feature which leads to GS.

### 3.3 Gradient Starvation Regime

In general, we do not expect to find an analytical solution for the dynamics of the coupled non-linear dynamical system of Eq. 10. However, there are at least two cases where a decoupled form for the dynamics allows to find an exact solution. We first introduce these cases and then study their perturbation to outline general lessons.

1. If the matrix of singular values $\mathbf{S}^2$ is proportional to the identity: This is the case where all the features have the same strength $s^2$. The fixed points are then given by,

$$\alpha_i^* = \frac{\lambda \mathcal{W}(\lambda^{-1}s^2 + 1)}{s^2 + \lambda}, \qquad z_j^* = \frac{s^2 \mathcal{W}(\lambda^{-1}s^2 + 1)}{s^2 + \lambda}\sum_i u_{ij}, \tag{12}$$

where $\mathcal{W}$ is the Lambert W function.

2. If the matrix $\mathbf{U}$ is a permutation matrix: This is the case in which each feature is associated with a single example only. The fixed points are then given by,

$$\alpha_i^* = \frac{\lambda \mathcal{W}(\lambda^{-1}s_i^2 + 1)}{s_i^2 + \lambda}, \qquad z_j^* = \frac{s_i^2 \mathcal{W}(\lambda^{-1}s_i^2 + 1)}{s_i^2 + \lambda}. \tag{13}$$

To study a minimal case of starvation, we consider a variation of case 2 with the following assumption which implies that each feature is not associated with a single example anymore.

**Lemma 1.** *Assume $\mathbf{U}$ is a perturbed identity matrix (a special case of a permutation matrix) in which the off-diagonal elements are proportional to a small parameter $\delta > 0$. Then, the fixed point of the dynamical system in Eq. 10 can be approximated by,*

$$\boldsymbol{\alpha}^* = (1 - \log(\boldsymbol{\alpha}_0^*))\left[\mathbf{A} + diag\left(\boldsymbol{\alpha}_0^{*-1}\right)\right]^{-1}, \tag{14}$$

*where $\mathbf{A} = \lambda^{-1}\mathbf{U}(\mathbf{S}^2 + \lambda\mathbf{I})\mathbf{U}^T$ and $\boldsymbol{\alpha}_0^*$ is the fixed point of the uncoupled system with $\delta = 0$.*

For sake of ease of derivations, we consider the two dimensional case where,

$$\mathbf{U} = \begin{pmatrix} \sqrt{1 - \delta^2} & -\delta \\ \delta & \sqrt{1 - \delta^2} \end{pmatrix}, \tag{15}$$

which is equivalent to a $U$ matrix with two blocks of features with no intra-block coupling and $\delta$ amount of inter-block coupling.

**Theorem 2** (Gradient Starvation Regime). *Consider a neural network in the linear regime, trained under cross-entropy loss for a binary classification task. With definition 1, assuming coupling between features 1 and 2 as in Eq. 15 and $s_1^2 > s_2^2$, we have,*

$$\frac{\mathrm{d}z_2^*}{\mathrm{d}s_1^2} < 0, \tag{16}$$

*which implies GS.*

While Thm. 2 outlines conditions for GS in two dimensional feature space, we note that the same rationale naturally extends to higher dimensions, where GS is defined pairwise over feature directions. For a classification task, Thm. 2 indicates that gradient starvation occurs when the data admits different feature strengths, and coupled learning dynamics. GS is thus naturally expected with cross-entropy loss. Its detrimental effects however (as outlined in Sect. 2) arise in settings with large discrepancies between feature strengths, along with network connectivity that couples these features' directions. This phenomenon readily extends to multi-class settings, and we validate this case with experiments in Sect. 4. Next, we introduce a simple regularizer that encourages feature decoupling, thus mitigating GS by insulating strong features from weaker ones.

### 3.4 Spectral Decoupling

By tracing back the equations of the previous section, one may realize that the term $U^T S^2 U$ in Eq. 9 is not diagonal in the general case, and consequently introduces coupling between $\alpha_i$'s and hence, between the features $z_i$'s. We would like to discourage solutions that couple features in this way. To that end, we introduce a simple regularizer: Spectral Decoupling (SD). SD replaces the general L2 weight decay term in Eq. 5 with an L2 penalty exclusively on the network's logits, yielding

$$\mathcal{L}\left(\boldsymbol{\theta}\right) = \mathbf{1} \cdot \log\left[1 + \exp\left(-\mathbf{Y}\hat{\mathbf{y}}\right)\right] + \frac{\lambda}{2}\|\hat{\mathbf{y}}\|^2. \tag{17}$$

Repeating the same analysis steps taken above, but with SD instead of general L2 penalty, the critical value for $\boldsymbol{\theta}^*$ becomes $\boldsymbol{\theta}^* = \frac{1}{\lambda}\boldsymbol{\alpha}\mathbf{Y}\boldsymbol{\Phi}_0 V\mathbf{S}^{-2}V^T$. This new expression for $\boldsymbol{\theta}^*$ results in the following modification of Eq. 9,

$$\dot{\boldsymbol{\alpha}} = \eta\left(\log\frac{\mathbf{1}-\boldsymbol{\alpha}}{\boldsymbol{\alpha}} - \frac{1}{\lambda}\boldsymbol{\alpha}\mathbf{U}\mathbf{S}^2\mathbf{S}^{-2}\mathbf{U}^T\right) = \eta\left(\log\frac{\mathbf{1}-\boldsymbol{\alpha}}{\boldsymbol{\alpha}} - \frac{1}{\lambda}\boldsymbol{\alpha}\right), \tag{18}$$

where as earlier, $\log$ and division are taken element-wise on the coordinates of $\boldsymbol{\alpha}$.

Note that in contrast to Eq. 9 the matrix multiplication involving $U$ and $S$ in Eq. 18 cancels out, leaving $\alpha_i$ independent of other $\alpha_{j\neq i}$'s. We point out this is true for any initial coupling, without simplifying assumptions. Thus, a simple penalty on output weights promotes decoupled dynamics across the dual parameter $\alpha_i$'s, which track learning dynamics of feature responses (see Eq. 7). Together with Thm. 2, Eq. 18 suggests SD should mitigate GS and promote balanced learning dynamics across features. We now verify this in numerical experiments. For further intuition, we provide a simple experiment, summarized in Fig. 5, where directly visualizes the primal vs. the dual dynamics as well as the effect of the proposed spectral decoupling method.

## 4 Experiments

The experiments presented here are designed to outline the presence of GS and its consequences, as well as the efficacy of our proposed regularization method to alleviate them. Consequently, we highlight that achieving state-of-the-art results is not the objective. For more details including the scheme for hyper-parameter tuning, see App. B.

### 4.1 Two-Moon classification and the margin

Recall the simple 2-D classification task between red and blue data points in Fig. 1. Fig. 1 **(c)** demonstrates the learned decision boundary when SD is used. SD leads to learning a curved decision boundary with a larger margin in the input space. See App. B for additional details and experiments.

### 4.2 CIFAR classification and adversarial robustness

To study the classification margin in deeper networks, we conduct a classification experiment on CIFAR-10, CIFAR-100, and CIFAR-2 (cats vs dogs of CIFAR-10) [57] using a convolutional network with ReLU non-linearity. Unlike linear models, the margin to a non-linear decision boundary cannot be computed analytically. Therefore, following the approach in [72], we use "the norm of input-disturbance required to cross the decision boundary" as a proxy for the margin. The disturbance on the input is computed by projected gradient descent (PGD) [84], a well-known adversarial attack.

| Dataset | Method | Train* | Test IID | Test OOD† |
|---|---|---|---|---|
| Cifar-2 | w/o SD | 100.0% ± 0.0 | 95.2% ± 0.12 | 42.3% ± 3.0 |
| | w/ SD$_{(\lambda=0.01)}$ | 100.0% ± 0.0 | 95.3% ± 0.17 | 69.7% ± 2.9 |
| Cifar-10 | w/o SD | 99.9% ± 0.01 | 92.8% ± 0.15 | 30.1% ± 2.1 |
| | w/ SD$_{(\lambda=0.01)}$ | 99.9% ± 0.01 | 92.9% ± 0.16 | 67.7% ± 1.5 |
| Cifar-100 | w/o SD | 99.7% ± 0.01 | 69.2% ± 0.29 | 14.3% ± 2.0 |
| | w/ SD$_{(\lambda=0.05)}$ | 99.7% ± 0.02 | 70.5% ± 0.26 | 24.9% ± 1.9 |

† Accuracy (± std) for 10 runs.

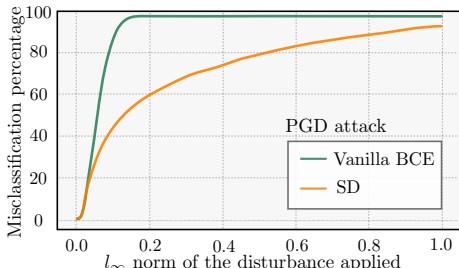

Table 1: Table compares adversarial robustness of ERM (vanilla cross-entropy) vs SD with a CNN trained on CIFAR-2, 10, and 100 (setup of [72]). SD consistently achieves a better OOD performance.

Figure 2: The plot shows the cumulative distribution function (CDF) of the margin for the CIFAR-2 binary classification. SD appears to improve the margin considerably.

Table 1 includes the results for IID (original test set) and OOD (perturbed test set by $\epsilon_{\mathrm{PGD}} = 0.05$). Fig. 2 shows the percentage of mis-classifications as the norm of disturbance is increased for the Cifar-2 dataset. This plot can be interpreted as the cumulative distribution function (CDF) of the margin and hence a lower curve reads as a more robust network with a larger margin. This experiment suggests that when trained with vanilla cross-entropy, even slight disturbances in the input deteriorates the network's classification accuracy. That is while spectral decoupling (SD) improves the margin considerably. Importantly, this improvement in robustness does not seem to compromise the noise-free test performance. It should also be highlighted that SD does not explicitly aim at maximizing the margin and the observed improvement is in fact a by-product of decoupled learning of latent features. See Section 5 for a discussion on why cross-entropy results in a poor margin while being considered a max-margin classifier in the literature [94].

### 4.3 Colored MNIST with color bias

We conduct experiments on the Colored MNIST Dataset, proposed in [9]. The task is to predict binary labels $y = -1$ for digits 0 to 4 and $y = +1$ for digits 5 to 9. A color channel (red, green) is artificially added to each example to deliberately impose a spurious correlation between the color and the label. The task has three environments:

- Training env. 1: Color is correlated with the labels with 0.9 probability.
- Training env. 2: Color is correlated with the labels with 0.8 probability.
- Testing env.: Color is correlated with the labels with 0.1 probability (0.9 reversely correlated).

Because of the opposite correlation between the color and the label in the test set, only learning to classify based on color would be disastrous at testing. For this reason, Empirical Risk Minimization (ERM) performs very poorly on the test set (23.7 % accuracy) as shown in Tab. 2.

| Method | Train Accuracy | Test Accuracy |
|---|---|---|
| **ERM** (Vanilla Cross-Entropy) | 91.1 % ($\pm$0.4) | 23.7 % ($\pm$0.8) |
| **REx** [59] | 71.5 % ($\pm$1.0) | 68.7 % ($\pm$0.9) |
| **IRM** [9] | 70.5 % ($\pm$0.6) | 67.1 % ($\pm$1.4) |
| **SD** (this work) | 70.0 % ($\pm$0.9) | 68.4 % ($\pm$1.2) |
| **Oracle** - (grayscale images) | 73.5 % ($\pm$0.2) | 73.0 % ($\pm$0.4) |
| **Random Guess** | 50 % | 50 % |

Table 2: Test accuracy on test examples of the Colored MNIST after training for 1k epochs. The standard deviation over 10 runs is reported in parenthesis. ERM stands for the empirical risk minimization. Oracle is an ERM trained on grayscale images. Note that due to 25 % label noise, a hypothetical optimum achieves 75 % accuracy (the upper bound).

Invariant Risk Minimization (IRM) [9] on the other hand, performs well on the test set with (67.1 % accuracy). However, IRM requires access to multiple (two in this case) separate training environments with varying amount of spurious correlations. IRM uses the variance between environments as a signal for learning to be "invariant" to spurious correlations. Risk Extrapolation (REx) [59] is a related training method that encourages learning invariant representations. Similar to IRM, it requires access to multiple training environments in order to quantify the concept of "invariance".

SD achieves an accuracy of 68.4 %. Its performance is remarkable because unlike IRM and REx, SD does not require access to multiple environments and yet performs well when trained on a single environment (in this case the aggregation of both of the training environments).

A natural question that arises is **"How does SD learn to ignore the color feature without having access to multiple environments?"** The short answer is that **it does not**! In fact, we argue that SD learns the color feature but it **also** learns other predictive features, i.e., the digit shape features. At test time, the predictions resulting from the shape features prevail over the color feature. To validate this hypothesis, we study a trained model with each of these methods (ERM, IRM, SD) on four variants of the test environment: 1) grayscale-digits: No color channel is provided and the network should rely on shape features only. 2) colored-digits: Both color and digit are provided however the color is negatively correlated (opposite of the training set) with the label. 3) grayscale-blank: All images are grayscale and blank and hence do not provide any information. 4) colored-blank: Digit features are removed and only the color feature is kept, also with reverse label compared to training. Fig. 3 summarizes the results. For more discussions see SM B.

As a final remark, we should highlight that, by design, this task assumes access to the test environment for hyperparameter tuning for all the reported methods. This is not a valid assumption in general, and

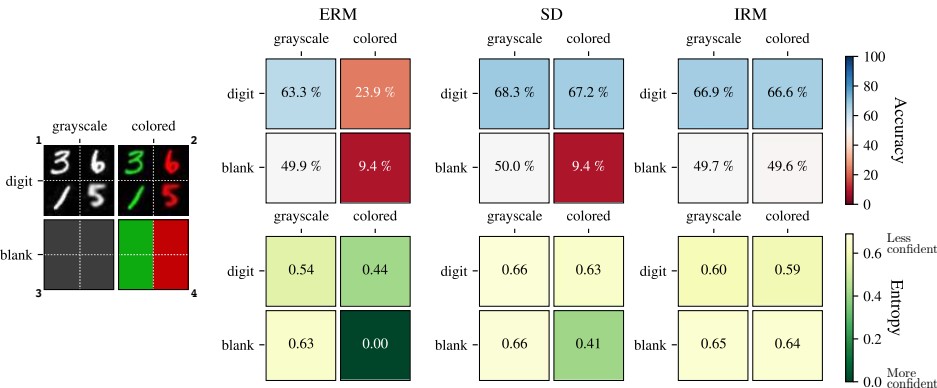

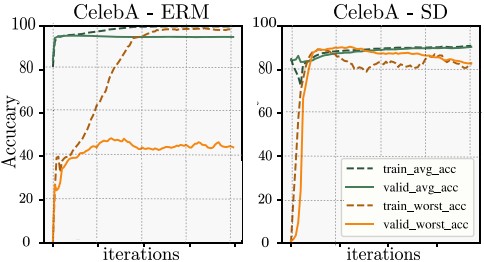

Figure 3: Diagram comparing ERM, SD, and IRM on four different test environments on which we evaluate a pre-trained model. Top and bottom rows show the accuracy and the entropy (inverse of confidence), respectively. **Analysis:** Compare three values of  9.4 % ,  9.4 % , and  49.6 % : Both ERM and SD have learned the color feature but since it is inversely correlated with the label, when only the color feature is provided, as expected both ERM and SD performs poorly. Now compare  0.00  and  0.41 : Although both ERM and SD have learned the color feature, ERM is much more confident on its predictions (zero entropy). As a consequence, when digit features are provided along with the color feature (colored-digit environment), ERM still performs poorly ( 23.9 % ) but SD achieves significantly better results ( 67.2 % ). IRM ignores the color feature altogether but it requires access to multiple training environments.

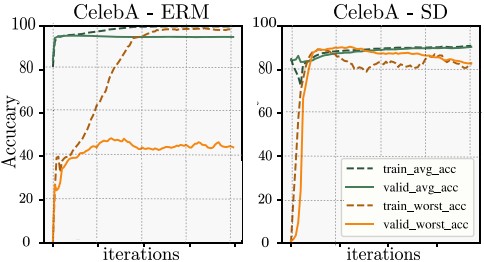

| Method | Average Acc. | Worst Group Acc. |
|---|---|---|
| **ERM** | 94.61 % ($\pm$0.67) | 40.35 % ($\pm$1.68) |
| **SD** (this work) | 91.64 % ($\pm$0.61) | **83.24** % ($\pm$2.01) |
| **LfF** | N/A | 81.24 % ($\pm$1.38) |
| **Group DRO**$^*$ | 91.76 % ($\pm$0.28) | 87.78 % ($\pm$0.96) |

Figure 4: CelebA: `blond` vs `dark` hair classification. The `HairColor` and the `Gender` are spuriously correlated which leads to poor OOD performance with ERM, however SD significantly improves performance. ERM's worst group accuracy is significantly lower than SD.

Table 3: CelebA: `blond` vs `dark` hair classification with spurious correlation. We report test performance over ten runs. SD significantly improves upon ERM. $^*$Group DRO [89] requires explicit information about the spurious correlation. LfF [71] requires simultaneous training of two networks.

hence the results should be only interpreted as a probe that shows that SD could provide an important level of control over what features are learned.

## 4.4 CelebA with gender bias

The CelebA dataset [65] contains 162k celebrity faces with binary attributes associated with each image. Following the setup of [89], the task is to classify images with respect to their hair color into two classes of blond or dark hair. However, the `Gender` $\in$ {`Male`, `Female`} is spuriously correlated with the `HairColor` $\in$ {`Blond`, `Dark`} in the training data. The rarest group which is blond males represents only 0.85 % of the training data (1387 out of 162k examples). We train a ResNet-50 model [38] on this task. Tab. 3 summarizes the results and compares the performance of several methods. A model with vanilla cross-entropy (ERM) appears to generalize well on average but fails to generalize to the rarest group (blond males) which can be considered as "weakly" out-of-distribution (OOD). Our proposed SD improves the performance more than twofold. It should be highlighted that for this task, we use a variant of SD in which, $\frac{\lambda}{2}||\hat{y} - \gamma||_2^2$ is added to the original cross-entropy loss. The hyper-parameters $\lambda$ and $\gamma$ are tuned separately for each class (a total of four hyper-parameters). This variant of SD does provably decouple the dynamics too but appears to perform better than the original SD in Eq. 17 in this task.

Other proposed methods presented in Tab. 3 also show significant improvements on the performance

of the worst group accuracy. The recently proposed "Learning from failure" (LfF) [71] achieves comparable results to SD, but it requires simultaneous training of two networks. Group DRO [89] is another successful method for this task. However, unlike SD, Group DRO requires explicit information about the spuriously correlated attributes. In most practical tasks, information about the spurious correlations is not provided and, dependence on the spurious correlation goes unrecognized.[2]

## 5  Related Work and Discussion

Here, we discuss the related work. Due to space constraints, further discussions are in App. A.

**On learning dynamics and Loss Choice.**  Several works including [90, 91, 1, 60] investigate the dynamics of deep linear networks trained with squared-error loss. Different decompositions of the learning process for neural networks have been used: [83, 104, 87, 105] study the learning in the Fourier domain and show that low-frequency functions are learned earlier than high-frequency ones. [90, 2, 32] provide closed-form equations for the dynamics of linear networks in terms of the principal components of the input covariance matrix. More recently, with the introduction of neural tangent kernel (NTK) [52, 62], a new line of research is to study the convergence properties of gradient descent [e.g. 8, 69, 25, 29, 7, 44, 33, 110, 11, 99]. Among them, [12, 106, 18, 22] decompose the learning process along the principal components of the NTK. The message in these works is that the training process can be decomposed into independent learning dynamics along the orthogonal directions.

Most of the studies mentioned above focus on the particular squared-error loss. For a linearized network, the squared-error loss results in linear learning dynamics, which often admit an analytical solution. However, the de-facto loss function for many of the practical applications of neural networks is the cross-entropy. Using the cross-entropy as the loss function leads to significantly more complicated and non-linear dynamics, even for a linear neural network. In this work, our focus was the cross-entropy loss.

**On reliance upon spurious correlations and robustness.**  In the context of robustness in neural networks, state-of-the-art neural networks appear to naturally focus on low-level superficial correlations rather than more abstract and robustly informative features of interest (e.g. [30]). As we argue in this work, Gradient Starvation is likely an important factor contributing to this phenomenon and can result in adversarial vulnerability. There is a rich research literature on adversarial attacks and neural networks' vulnerability [96, 34, 48, 67, 5, 47]. Interestingly, [73], [72] and [51] draw a similar conclusion and argue that "an insufficiency of the cross-entropy loss" causes excessive invariances to predictive features. Perhaps [92] is the closest to our work in which authors study the simplicity bias (SB) in stochastic gradient descent. They demonstrate that neural networks exhibit extreme bias that could lead to adversarial vulnerability.

**On implicit bias.**  Despite being highly-overparameterized, modern neural networks seem to generalize very well [108]. Modern neural networks generalize surprisingly well in numerous machine tasks. This is despite the fact that neural networks typically contain orders of magnitude more parameters than the number of examples in a training set and have sufficient capacity to fit a totally randomized dataset perfectly [108]. The widespread explanation is that the gradient descent has a form of implicit bias towards learning simpler functions that generalize better according to Occam's razor. Our exposition of GS reinforces this explanation. In essence, when training and test data points are drawn from the same distribution, the top salient features are predictive in both sets. We conjecture that in such a scenario, by not learning the less salient features, GS naturally protects the network from overfitting.

The same phenomenon is referred to as *implicit bias*, *implicit regularization*, *simplicity bias* and *spectral bias* in several works [83, 75, 36, 74, 70, 53, 94, 10, 13, 35, 82, 66].

As an active line of research, numerous studies have provided different explanations for this phenomenon. For example, [70] justifies the implicit bias of neural networks by showing that stochastic gradient descent learns simpler functions first. [15, 78] suggests that a form of implicit regularization is induced by an alignment between NTK's principal components and only a few task-relevant

---

[2]Recall that it took 3 years for the psychologist, Oskar Pfungst, to realize that Clever Hans was not capable of doing any arithmetic.

directions. Several other works such as [20, 35, 94, 25] recognize the convergence of gradient descent to maximum-margin solution as the essential factor for the generalizability of neural networks. It should be stressed that these work refer to the margin in the hidden space and not in the input space as pointed out in [55]. Indeed, as observed in our experiments, the maximum-margin classifier in the hidden space can be achieved at the expense of a small margin in the input space.

**On Gradient Starvation and no free lunch theorem.** The *no free lunch* theorem [93, 102] states that "learning is impossible without making assumptions about training and test distributions". Perhaps, the most commonly used assumption of machine learning is the i.i.d. assumption [98], which assumes that training and test data are identically distributed. However, in general, this assumption might not hold, and in many practical applications, there are predictive features in the training set that do not generalize to the test set. A natural question that arises is *how to favor generalizable features over spurious features?* The most common approaches include *data augmentation, controlling the inductive biases, using regularizations,* and more recently *training using multiple environments*.

Here, we would like to elaborate on an interesting thought experiment of [79]: Suppose a neural network is provided with a chess book containing examples of chess games with the best movements indicated by a red arrow. The network can take two approaches: 1) learn how to play chess, or 2) learn just the red arrows. Either of these solutions results in zero training loss on the games in the book while only the former is generalizable to new games. With no external knowledge, the network typically learns the simpler solution.

Recent work aims to leverage the invariance principle across several environments to improve robust learning. This is akin to present several chess books to a network, each with markings indicating the best moves for different sets of games. In several studies [9, 59, 79, 3], methods are developed to aggregate information from multiple training environments in a way that favors the generalizable / domain-agnostic / invariant solution. We argue that even with having access to **only one** training environment, there is useful information in the training set that fails to be discovered due to Gradient Starvation. The information on how to actually play chess is already available in any of the chess books. Still, as soon as the network learns the red arrows, the network has no incentive for further learning. Therefore, *learning the red arrows is not an issue per se, but not learning to play chess is.*

**Gradient Starvation: friend or foe?** Here, we would like to remind the reader that GS can have both adverse and beneficial consequences. If the learned features are sufficient to generalize to the test data, gradient starvation can be viewed as an implicit regularizer. Otherwise, Gradient Starvation could have an unfavorable effect, which we observe empirically when some predictive features fail to be learned. A better understanding and control of Gradient Starvation and its impact on generalization offers promising avenues to address this issue with minimal assumptions. Indeed, our Spectral Decoupling method requires an assumption about feature imbalance but not to pinpoint them exactly, relying on modulated learning dynamics to achieve balance.

**GS social impact** Modern neural networks are being deployed extensively in numerous machine learning tasks. Our models are used in critical applications such as autonomous driving, medical prediction, and even justice system where human lives are at stake. However, neural networks appear to base their predictions on superficial biases in the dataset. Unfortunately, biases in datasets could be neglected and pose negative impacts on our society. In fact, our Celeb-A experiment is an example of the existence of such a bias in the data. As shown in the paper, the gender-specific bias could lead to a superficial high performance and is indeed very hard to detect. Our analysis, although mostly on the theory side, could pave the path for researchers to build machine learning systems that are robust to biases and helps towards fairness in our predictions.

## 6   Conclusion

In this paper, we formalized Gradient Starvation (GS) as a phenomenon that emerges when training with cross-entropy loss in neural networks. By analyzing the dynamical system corresponding to the learning process in a dual space, we showed that GS could slow down the learning of certain features, even if they are present in the training set. We derived spectral decoupling (SD) regularization as a possible remedy to GS.

## Acknowledgments and Disclosure of Funding

The authors are grateful to Samsung Electronics Co., Ldt., CIFAR, and IVADO for their funding and Calcul Québec and Compute Canada for providing us with the computing resources. We would further like to acknowledge the significance of discussions and supports from Reyhane Askari Hemmat and Faruk Ahmed. MP would like to thank Aristide Baratin, Kostiantyn Lapchevskyi, Seyed Mohammad Mehdi Ahmadpanah, Milad Aghajohari, Kartik Ahuja, Shagun Sodhani, and Emmanuel Bengio for their invaluable help.

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
