# A  Further discussions

**On Primal (parameter space) vs. Dual (feature space) dynamics:** Although the cross-entropy loss is convex, it does not admit an analytical solution, even in a simple logistic regression [103]. Importantly, it also does not have a finite solution when the data is linearly separable [6] (which is the case in high dimensions [27]). As such, our study is concerned with characterizing the solutions that the training algorithm converges to. A dual optimization approach enables us to describe these solutions in terms of contributions of the training examples [43]. While primal and dual dynamics are not guaranteed to match, the solution they converge to is guaranteed to match [21], and that is what our theory builds upon.

For further intuition, we provide a simple experiment in app C, directly visualizing the primal vs. the dual dynamics as well as the effect of the proposed spectral decoupling method.

**The intuition behind Spectral Decoupling (SD):** Consider a training datapoint $x$ in the middle of the training process. Intuitively, the model has two options for decreasing the loss of this example:

1. Get more confident on a feature that has been learned already by other examples. or,
2. Learn a new feature.

SD, a simple L2 penalty on the output of the work, would favor (2) over (1). The reason is that (2) does not make the network over-confident on previously learned examples, while (1) results in over-confident predictions. Hence, SD encourages learning more features by penalizing confidence. Our principal novel contribution is to characterize this process formally and to theoretically and empirically demonstrate its effectiveness.

From another perspective, here we describe how one can arrive at Spectral Decoupling. From Thm. 2, we know that Gradient Starvation happens because of the coupling between features (equivalently alphas). We notice that in Eq. 9, if we get rid of $S^2$, then the alphas are decoupled. To get rid of $S^2$, one can see that instead of $||\boldsymbol{\theta}||^2$ as the regularizer, we should have $||SV^T\boldsymbol{\theta}||^2$. Luckily, this is exactly equal to $||\hat{y}^2||$, since $\hat{y} = \Phi\boldsymbol{\theta} = UV^T\boldsymbol{\theta}$. We would like to highlight that $||SV^T\boldsymbol{\theta}||^2$ as the regularizer means that different directions are penalized according to their strength. It means that we suppress stronger directions more than others which would allow weaker directions to flourish.

**Then why not use Squared-error loss for classification too?** The biggest obstacle when using squared-error loss for classification is how to select the target. For example, in a cats vs. dogs classification task, not all cats have the same amount of "catty features". However, recent results favor using squared-error loss for classification and show that models trained with squared-error loss are more robust [46]. We conjecture that the improved robustness can be attributed to a lack of gradient starvation.

**On using NTK:** Theoretical analysis of neural networks in their general form is challenging and generally intractable. Neural Tangent Kernel (NTK) has been an important milestone that has simplified theoretical analysis significantly and provides some mechanistic explanations that are applicable in practice. Inevitably, it imposes a set of restrictions; mainly, NTK is only accurate in the limit of large width. Therefore, the common practice is to provide the theoretical analysis in simplified settings and validate the results empirically in more general cases (see, e.g. [45, 24, 101]). In this work, we build on the same established practices: Our theories analytically study an NTK linearized network; and we further validate our findings on several standard neural networks. In fact, in all of our experiments, learning is done in the regular "rich" (non-NTK) regime, and we verify that our proposed method, as identified analytically, mitigates learning limitations.

**Future Directions:** This work takes a step towards understanding the reliance of neural networks upon spurious correlations and shortcuts in the dataset. We believe identifying this reliance in sensitive applications is among the next steps for future research directions. That would have a pronounced real-world impact as neural networks have started to be used in many critical applications. As a recent example, we would like to point to an article by researchers at Cambridge [86] where they study more than 300 papers on detecting whether a patient has COVID or not given their CT Scans. According to the article, none of the papers were able to generalize from one hospital data to another since the models learn to latch on to hospital-specific features. An essential first step is to uncover such reliance and then to design methods such as our proposed spectral decoupling to mitigate the problem.

# B  Experimental Details

## B.1  A Simple Experiment Summarizing the Theory

Here, we provide a simple experiment to study the difference between the primal and dual form dynamics. We also compare the learning dynamics in cases with and without Spectral Decoupling (SD).

Recall that primal dynamics arise from the following optimization,

$$\min_{\theta} \left( \mathbf{1} \cdot \log\left[1 + \exp\left(-\mathbf{Y}\hat{\mathbf{y}}\right)\right] + \frac{\lambda}{2}\left\|\hat{\boldsymbol{\theta}}\right\|^2 \right),$$

while the dual dynamics are the result of another optimization,

$$\max_{\alpha_i} \left[ H(\boldsymbol{\alpha}) - \frac{1}{2\lambda}\sum_{jq} \boldsymbol{\alpha}\mathbf{Y}\boldsymbol{\Phi}_0\boldsymbol{\Phi}_0^T\mathbf{Y}^T\boldsymbol{\alpha}^T \right].$$

Also recall that Spectral Decoupling suggests the following optimization,

$$\min_{\theta} \left( \mathbf{1} \cdot \log\left[1 + \exp\left(-\mathbf{Y}\hat{\mathbf{y}}\right)\right] + \frac{\lambda}{2}\|\hat{\mathbf{y}}\|^2 \right).$$

We conduct experiments on a simple toy classification with two datapoints for which the matrix $\mathbf{U}$ of Eq. 15 is defined as, $\mathbf{U} = \begin{pmatrix} 0.8 & -0.6 \\ 0.6 & 0.8 \end{pmatrix}$. The corresponding singular values $\mathbf{S} = [s_1, s_2 = 2]$ where $s_1 \in \{2, 3, 4, 5, 6\}$. According to Eq. 13, when $\mathbf{S} = [2, 2]$, the dynamics decouple while in other cases starvation occurs. Fig. 5 shows the corresponding features of $z_1$ and $z_2$. It is evident that by increasing the value of $s_1$, the value of $z_1^*$ increases while $z_2^*$ decreases (starves). Fig. 5 (left) also compares the difference between the primal and the dual dynamics. Note that although their dynamics are different, they both share the same fixed points. Fig. 5 (right) also shows that Spectral Decoupling (SD) indeed decouples the learning dynamics of $z_1$ and $z_2$ and hence increasing the corresponding singular value of one does not affect the other.

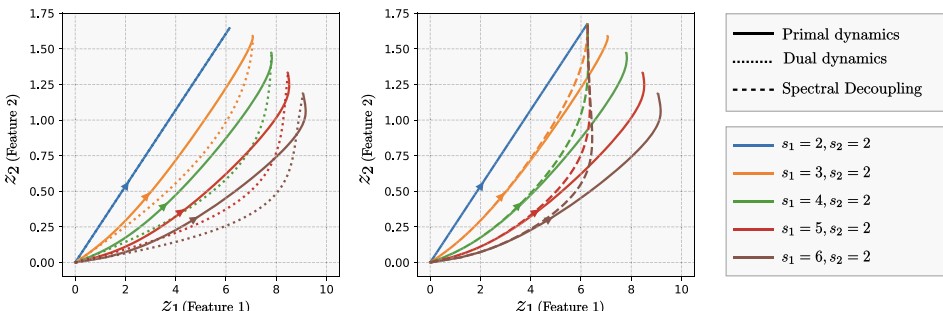

Figure 5: An illustration of the learning dynamics for a simple 2D classification task. x-axis and y-axis represent learning along features of $z_1$ and $z_2$, respectively. Each trajectory corresponds to a combination of the corresponding singular values of $s_1$ and $s_2$. It is evident that by increasing the value of $s_1$, the value of $z_1^*$ increases while $z_2^*$ decreases (starves). **(Left)** compares the difference between the primal and the dual dynamics. Note that although their dynamics are different, they both share the same fixed points. **(Right)** shows that Spectral Decoupling (SD) indeed decouples the learning dynamics of $z_1$ and $z_2$ and hence increasing the corresponding singular value of one does not affect the other.

## B.2  Two-Moon Classification: Comparison with other regularization methods

We experiment the Two-moon classification example of the main paper with different regularization techniques. The small margin between the two classes allows the network to achieve a negligible loss by only learning to discriminate along the horizontal axis. However, both axes are relevant for the data distribution, and the only reason why the second dimension is not picked up is the fact that the

training data allows the learning to explain the labels with only one feature, overlooking the other. Fig. 6 reveals that common regularization strategies including Weight Decay, Dropout [95] and Batch Normalization [49] do not help achieving a larger margin classifier. Unless states otherwise, all the methods are trained with Full batch Gradient Descent with a learning rate of $1e-2$ and a momentum of $0.9$ for $10k$ iterations.

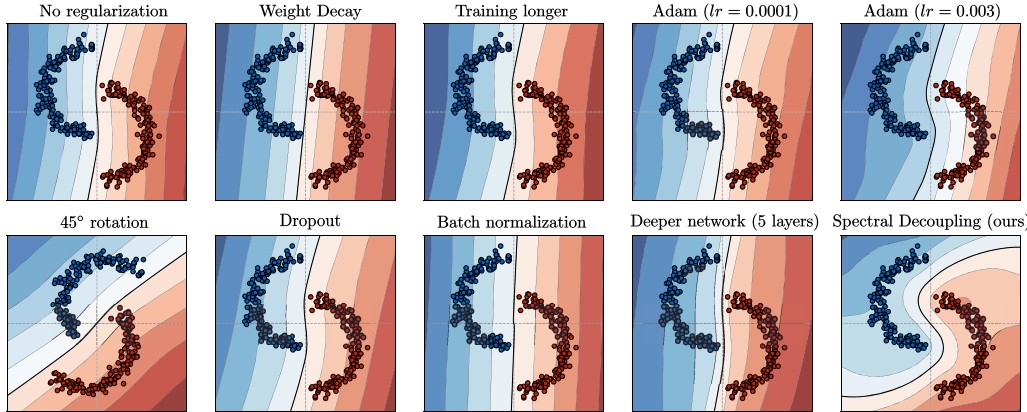

Figure 6: The effect of common regularization methods on a simple task of two-moon classification. It can be seen that common practices of deep learning seem not to help with learning a curved decision boundary. The acronym "lr" for the Adam optimizer refers to the learning rate. Shown decision boundaries are the average over 10 runs in which datapoints and the model initialization parameters are sampled randomly. Here, only the datapoints of one particular seed are plotted for visual clarity.

## B.3 CIFAR classification

We use a four-layer convolutional network with ReLU non-linearity following the exact setup of [72]. Sweeping $\lambda$ from 0 to its optimal value results in a smooth transition from green to orange. However, larger values of $\lambda$ will hurt the IID test (zero perturbation) generalization. The value that we cross-validate on is the average of IID and OOD generalization performance.

## B.4 Colored MNIST with color bias

For the Colored MNIST task, we aggregate all the examples from both training environments. Table. 4 reports the hyper-parameters used for each method.

| Method | Layers | Dim | Weight decay | LR | Anneal steps | Penalty coef |
|--------|--------|-----|--------------|-----|--------------|--------------|
| ERM | 2 | 300 | 0.0 | 1e-4 | 0/2000 | n/a |
| SD | 2 | 300 | 0.0 | 1e-4 | 450/2000 | 2e-5 |
| IRM | 2 | 390 | 0.00110794568 | 0.00048985365 | 190/500 | 91257.18613115903 |

Table 4: Hyper-parameters used for the Colored-MNIST experiment. Hyper-parameters of IRM are obtained from their released code. "Anneal steps" indicates the number of iterations done before applying the method.

**More on Fig. 3**. ERM captures the color feature and in the absence of any digit features (environment 4), the network's accuracy is low, as is expected because of reverse color-label match at testing. Moreover, the ERM network is very confident in this environment (confidence is inversely proportional to entropy). The SD network appears to capture the color feature too, with identical classification accuracy in environment 4, but much lower confidence which indicates the other features it expects to classify are absent. Consistent with this, in the case where both color and digit features are present (environment 2), SD achieves significantly better performance than ERM which is fooled by the flipped colors. This is again consistent with SD mitigating GS caused by the color feature onto the digit shape features. Meanwhile, IRM appears to not capture the color feature altogether. Specifically, when only the color is presented to a network trained with IRM, network predicts $50\%$ accuracy

with low confidence meaning that IRM is indeed "invariant" to the color as its name suggests. We note that further theoretical justifications are required to fully understand the underlying mechanisms in learning with spurious correlations.

**As a final remark**, we highlight that, by design, this task assumes access to the test environment for hyperparameter tuning for all the reported methods. This is not a valid assumption in general, and hence the results should be only interpreted as a probe that shows that SD could provide an important level of control over what features are learned.

The hyperparameter search has resulted in applying the SD at $450^{th}$ step. We observe that $450^{th}$ step is the step at which the traditional (in-distribution) overfitting occurs. This suggests that one might be able to tune hyperparameters without the need to monitor on the test set.

For all the experiments, we use PyTorch [80]. We also use NNGeometry [31] for computing NTK.

### B.5 CelebA with gender bias: The experimental details

Figure 4 depicts the learning curves for this task with and without Spectral Decoupling. For the CelebA experiment, we follow the same setup as in [89] and use their released code. We use Adam optimizer for the Spectral Decoupling experiments with a learning rate of $1e - 4$ and a batch size of 128. As mentioned in the main text, for this experiment, we use a different variant of Spectral Decoupling which also provably decouples the learning dynamics,

$$\min_{\theta} \mathbf{1} \cdot \left( \log\left[1 + \exp\left(-\mathbf{Y}\hat{\mathbf{y}}\right)\right] + \frac{\lambda}{2}\|\hat{\mathbf{y}} - \gamma\|^2 \right).$$

#### B.5.1 Hyper-parameters

We applied a hyper-parameter search on $\lambda$ and $\gamma$ for each of the classes separately. Therefore, a total of four hyper-parameters are found. For class zero, $\lambda_0 = 0.088$, $\gamma_0 = 0.44$ and for class one, $\lambda_1 = 0.012$, $\gamma_1 = 2.5$ are found to result in the best worst-group performance.

During the experiments, we found that for the CelebA dataset, classes are imbalanced: 10875 examples for class 0 and 1925 examples for class 1; meaning a ratio of 5.65. That is why we decided to penalize examples of each class separately with different coefficients. We also found that penalizing the outputs' distance to different values $\gamma_0$ and $\gamma_1$ helps the generalization. As stated in lines 842-844, the hyperparameter search results in the following values: 2.5 and 0.44.

### B.6 Computational Resources

For the experiments and hyper-parameter search an approximate number of 800 GPU-hours has been used. GPUs used for the experiments are NVIDIA-V100 mostly on internal cluster and partly on public cloud clusters.

## C  Proofs of the Theories and Lemmas

### C.1  Eq. 7 Legendre Transformation

Following [50], we derive the Legendre transformation of the Cross-Entropy (CE) loss function. Here, we reiterate this transformation as following,

**Lemma 2** (CE's Legendre transformation, adapted from Eq. 46 of [50]). *For a variational parameter $\alpha \in [0, 1]$, the following linear lower bound holds for the cross-entropy loss function,*

$$\mathcal{L}(\omega) := \log\left(1 + e^{-\omega}\right) \geq H(\alpha) - \alpha\omega, \tag{19}$$

*in which $\omega := y\hat{y}$ and $H(\alpha)$ is the Shannon's binary entropy. The equality holds for the critical value of $\alpha^* = -\nabla_\omega \mathcal{L}$, i.e., at the maximum of r.h.s. with respect to $\alpha$.*

*Proof.* The **Legendre** transformation converts a function $\mathcal{L}(\omega)$ to another function $g(\alpha)$ of conjugate variables $\alpha$, $\mathcal{L}(\omega) \to g(\alpha)$. The idea is to find the expression of the tangent line to $\mathcal{L}(\omega)$ at $\omega_0$ which is the first-order Taylor expansion of $\mathcal{L}(\omega)$,

$$t(\omega, \omega_0) = \mathcal{L}(\omega_0) + (\omega - \omega_0) \nabla_\omega \mathcal{L}|_{\omega = \omega_0}, \tag{20}$$

where $t(\omega, \omega_0)$ is the tangent line. According to the Legendre transformation, the function $\mathcal{L}(\omega)$ can be written as a function of the intercepts of tangent lines (where $\omega = 0$). Varying $\omega_0$ along the $x$-axis provides us with a general equation, representing the intercept as a function of $\omega$,

$$t(\omega = 0, \omega_0 = \omega) = \mathcal{L}(\omega) - \omega \nabla_\omega \mathcal{L}. \tag{21}$$

The cross-entropy loss function can be rewritten as a soft-plus function,

$$\mathcal{L}(\omega) = -\log \sigma(\omega) = \log\big(1 + e^{-\omega}\big), \tag{22}$$

in which $\omega := y\hat{y}$. Letting $\alpha := -\nabla_\omega \mathcal{L} = \sigma(-\omega)$ we have,

$$\omega = \log\left(\frac{1 - \alpha}{\alpha}\right), \tag{23}$$

which allows us to re-write the expression for the intercepts as a function of $\alpha$ (denoted by $g(\alpha)$),

$$g(\alpha) = \mathcal{L}(\omega) - \omega \nabla_\omega \mathcal{L} \tag{24}$$
$$= \mathcal{L}(\omega) + \alpha \omega \tag{25}$$
$$= -\alpha \log \alpha - (1 - \alpha) \log(1 - \alpha) \tag{26}$$
$$= H(\alpha), \tag{27}$$

where $H(\alpha)$ is the binary entropy function.

Now, since $\mathcal{L}$ is convex, a tangent line is always a lower bound and therefore at its maximum it touches the original function. Consequently, the original function can be recovered as follows,

$$\mathcal{L}(\omega) = \max_{0 \leq \alpha \leq 1} H(\alpha) - \alpha \omega. \tag{28}$$

Note that the lower bound in Eq. 19 is now a linear function of $\omega := y\hat{y}$ but at the expense of an additional maximization over the variational parameter $\alpha$. An illustration of the lower bound is depicted in Fig. 7. Also a comparison between the dual formulation of other common loss functions is provided in Table. 5.

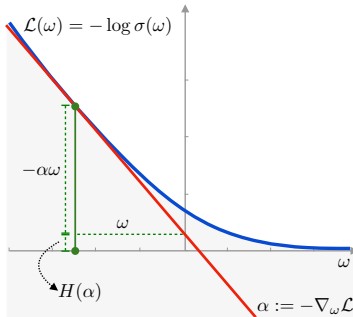

| Loss | Primal form | Dual form |
|---|---|---|
| **Cross-Entropy** | $\log(1 + e^{-w})$ | $\max_{0 < \alpha < 1} \big[H(\alpha) - \alpha \omega\big]$ |
| **Hinge loss** | $\max(0, 1 - w)$ | $\max_{0 \leq \alpha \leq 1} \big[\alpha \mathbf{1}^T - \alpha \omega\big]$ |
| **Squared error** | $(1 - \omega)^2$ | $\max_\alpha \big[-\frac{1}{2}\alpha^2 + \alpha - \alpha \omega\big]$ |

Figure 7: Diagram illustrating the Legendre transformation of the function $\mathcal{L}(\omega) = \log\big(1 + e^{-\omega}\big)$. The function is shown in blue, and the tangent line is shown in red. The tangent line is the lower bound of the function: $H(\alpha) - \alpha\omega \leq \mathcal{L}(\omega)$.

Table 5: Dual forms of other common different loss functions. The dual form of the Hinge loss is commonly used in Support Vector Machine (SVMs). For the ease of notation, we assume scalar $\omega$ and $\alpha$.

$\square$

### C.1.1 Extension to Multi-Class

Building on Eq. 65-71 of [50], which derives the Legendre transform of multi-class cross-entropy, one can update Eq. 6 of the main paper to

$$-\sum_{c=1}^{C} y_c \log(\hat{y}_c) \geq H(\alpha) - \sum_{c=1}^{C} \alpha^c \hat{y}_c, \tag{29}$$

where $H(\alpha)$ is the entropy function, $C = \#$classes, and vectors of $\alpha^c$ are defined for each class. Then Eq. 8 of the paper is then updated to,

$$\left( H(\alpha) - \frac{1}{2\lambda} \sum_{c=1}^{C} (\delta_y - \alpha^c) \Phi \Phi^T (\delta_y - \alpha^c)^T \right). \tag{30}$$

With a change of variable $\alpha^c := \delta_y - \alpha^c$, the theory of SD should remain unchanged.

### C.2 Eq. 8 Dual Dynamics

In Eq. 7, the order of min and max can be swapped as proved in Lemma 3 of [50], leading to,

$$\min_{\boldsymbol{\theta}} \mathcal{L}(\theta) = \max_{\boldsymbol{\alpha}} \min_{\boldsymbol{\theta}} \left( \mathbf{1} \cdot H(\boldsymbol{\alpha}) - \boldsymbol{\alpha} \mathbf{Y} \hat{\mathbf{y}} + \frac{\lambda}{2} \|\boldsymbol{\theta}\|^2 \right).$$

The solution to the inner optimization is,

$$\boldsymbol{\theta}^{*T} = \frac{1}{\lambda} \boldsymbol{\alpha} \mathbf{Y} \boldsymbol{\Phi}_0,$$

which its substitution into Eq. 7 results in Eq. 8.

### C.3 Eq. 9

Simply taking the derivative of Eq. 8 will result in Eq. 9. When we introduce continuous gradient ascent, we must define a learning rate parameter. This term is conceptually equivalent to the learning rate in SGD, but in this continuous setting, it has no influence on the fixed point.

### C.4 Eq. 10 Approximate Dynamics

Approximating dynamics of Eq. 9 with a first order Taylor expansion around the origin of the second term, we obtain

$$\dot{\boldsymbol{\alpha}} \approx \eta \left( -\log \boldsymbol{\alpha} - \frac{1}{\lambda} \boldsymbol{\alpha} \mathbf{U} \left( \mathbf{S}^2 + \lambda \mathbf{I} \right) \mathbf{U}^T \right).$$

*Proof.* Starting from the exact dynamics at Eq. 9,

$$\dot{\boldsymbol{\alpha}} = \eta \left( -\log \boldsymbol{\alpha} + \log \left( \mathbf{1} - \boldsymbol{\alpha} \right) - \frac{1}{\lambda} \boldsymbol{\alpha} \mathbf{U} \mathbf{S}^2 \mathbf{U}^T \right), \tag{31}$$

we perform a first-order Taylor approximation of the second term at $\boldsymbol{\alpha} = \mathbf{0}$ :

$$\log \left( \mathbf{1} - \boldsymbol{\alpha} \right) = -\boldsymbol{\alpha} + \mathcal{O} \left( \boldsymbol{\alpha}^2 \right). \tag{32}$$

Replacing in Eq. 31, we obtain

$$\dot{\boldsymbol{\alpha}} \approx \eta \left( -\log \boldsymbol{\alpha} - \boldsymbol{\alpha} - \frac{1}{\lambda} \boldsymbol{\alpha} \mathbf{U} \mathbf{S}^2 \mathbf{U}^T \right), \tag{33}$$

$$\dot{\boldsymbol{\alpha}} \approx \eta \left( -\log \boldsymbol{\alpha} - \boldsymbol{\alpha} \mathbf{U} \mathbf{I} \mathbf{U}^T - \frac{1}{\lambda} \boldsymbol{\alpha} \mathbf{U} \mathbf{S}^2 \mathbf{U}^T \right), \tag{34}$$

$$\dot{\boldsymbol{\alpha}} \approx \eta \left( -\log \boldsymbol{\alpha} - \frac{1}{\lambda} \boldsymbol{\alpha} \mathbf{U} \left( \mathbf{S}^2 + \lambda \mathbf{I} \right) \mathbf{U}^T \right). \tag{35}$$

$\square$

## C.5   Thm. 1 Attractive Fixed-Points

**Theorem.** *Any fixed points of the system in Eq. 10 is attractive in the domain $\alpha_i \in (0, 1)$.*

*Proof.* We define

$$f_i(\alpha_j) = \eta \left( \log \frac{(1 - \alpha_i)}{\alpha_i} - \frac{1}{\lambda} \sum_{jk} u_{ik} s_k^2 (u^T)_{kj} \alpha_j \right) \tag{36}$$

as the gradient function of the autonomous system Eq. 9.

We find the character of possible fixed points by linearization. We compute the jacobian of the gradient function evaluated at the fixed point.

$$J_{ik} = \left. \frac{\mathrm{d} f_i(\alpha_j)}{\mathrm{d} \alpha_k} \right|_{\alpha_k^*} \tag{37}$$

$$J_{ik} = \eta \left( -\delta_{ik} \left[ \alpha_i^* (1 - \alpha_i^*) \right]^{-1} - \lambda^{-1} \sum_l u_{il} s_l^2 (u^T)_{lk} \right) \tag{38}$$

The fixed point is an attractor if the jacobian is a negative-definite matrix. The first term is negative-definite matrix while the second term is negative semi-definite matrix. Since the sum of a negative matrix and negative-semi definite matrix is negative-definite, this completes the proof. $\square$

## C.6   Eq. 11 Feature Response at Fixed-Point

At the fixed point $\boldsymbol{\alpha}^*$, corresponding to the optimum of Eq. 8, the feature response of the neural network is given by,

$$\mathbf{z}^* = \frac{1}{\lambda} \mathbf{S}^2 \mathbf{U}^T \boldsymbol{\alpha}^{*T}.$$

*Proof.* The solution to the converged $\boldsymbol{\theta}^*$ at the Fixed-Point $\boldsymbol{\alpha}^*$ of Eq. 10 is,

$$\boldsymbol{\theta}^{*T} = \frac{1}{\lambda} \boldsymbol{\alpha}^* \mathbf{Y} \boldsymbol{\Phi}_0,$$

which by substitution into Eq. 4, Eq. 11 is derived.

$\square$

## C.7   Eq. 12 Uncoupled Case 1

If the matrix of singular values $\mathbf{S}^2$ is proportional to the identity, the fixed points of Eq. 10 are given by,

$$\alpha_i^* = \frac{\lambda \mathcal{W}(\lambda^{-1} s^2 + 1)}{s^2 + \lambda}, \qquad z_j^* = \frac{s^2 \mathcal{W}(\lambda^{-1} s^2 + 1)}{s^2 + \lambda} \sum_i u_{ij}, \tag{39}$$

where $\mathcal{W}$ is the Lambert W function.

*Proof.* When $\mathbf{S}^2 = s^2 \mathbf{I}$, Eq. 10 becomes

$$\dot{\boldsymbol{\alpha}} \approx \eta \left( -\log \boldsymbol{\alpha} - \frac{1}{\lambda} \boldsymbol{\alpha} \mathbf{U} \left( s^2 + \lambda \right) \mathbf{I} \mathbf{U}^T \right), \tag{40}$$

$$\dot{\boldsymbol{\alpha}} \approx \eta \left( -\log \boldsymbol{\alpha} - \frac{1}{\lambda} \boldsymbol{\alpha} \left( s^2 + \lambda \right) \right). \tag{41}$$

Fixed points of this system are obtained when $\dot{\boldsymbol{\alpha}} = 0$ :

$$0 = \eta \left( -\log \boldsymbol{\alpha} - \frac{s^2 + \lambda}{\lambda} \boldsymbol{\alpha} \right), \tag{42}$$

$$\log \boldsymbol{\alpha} = -\frac{s^2 + \lambda}{\lambda} \boldsymbol{\alpha}. \tag{43}$$

The solution of this equation is

$$\alpha_i^* = \frac{\lambda \mathcal{W}(\lambda^{-1}s^2 + 1)}{s^2 + \lambda} \tag{44}$$

With $\mathbf{z}$ given by Eq. 11, we have

$$\mathbf{z} = \frac{s^2 \mathcal{W}(\lambda^{-1}s^2 + 1)}{s^2 + \lambda}\mathbf{U}^T\mathbf{1}, \tag{45}$$

$$z_i^* = \frac{s^2 \mathcal{W}(\lambda^{-1}s^2 + 1)}{s^2 + \lambda}\sum_i u_{ij}. \tag{46}$$

$\square$

## C.8  Eq. 13 Uncoupled Case 2

If the matrix $\mathbf{U}$ is a permutation matrix, the fixed points of Eq. 10 are given by,

$$\alpha_i^* = \frac{\lambda \mathcal{W}(\lambda^{-1}s_i^2 + 1)}{s_i^2 + \lambda}, \qquad\qquad z_j^* = \frac{s_i^2 \mathcal{W}(\lambda^{-1}s_i^2 + 1)}{s_i^2 + \lambda}. \tag{47}$$

*Proof.* When $\mathbf{U}$ is a permutation matrix, it can be made an identity matrix with a meaningless reordering of the class labels . Without loss of generality, we therefore consider $\mathbf{U} = \mathbf{I}$

$$\dot{\boldsymbol{\alpha}} \approx \eta\left(-\log\boldsymbol{\alpha} - \frac{1}{\lambda}\boldsymbol{\alpha}\left(\mathbf{S}^2 + \lambda\right)\mathbf{I}\right). \tag{48}$$

Fixed points of this system are obtained when $\dot{\boldsymbol{\alpha}} = 0$

$$0 = \eta\left(-\log\alpha_i - \frac{1}{\lambda}\alpha_i\left(s_i^2 + \lambda\right)\right), \tag{49}$$

$$\log\alpha_i = -\frac{1}{\lambda}\alpha_i\left(s_i^2 + \lambda\right) \tag{50}$$

The solution of this equation is

$$\alpha_i^* = \frac{\lambda \mathcal{W}(\lambda^{-1}s_i^2 + 1)}{s_i^2 + \lambda}. \tag{51}$$

With $\mathbf{z}$ given by Eq. 11, we have

$$\mathbf{z}^* = \frac{1}{\lambda}\mathbf{S}^2\mathbf{I}\boldsymbol{\alpha}^{*T}, \tag{52}$$

$$z_j^* = \frac{s_i^2 \mathcal{W}(\lambda^{-1}s_i^2 + 1)}{s_i^2 + \lambda}. \tag{53}$$

$\square$

## C.9  Lemma 1 Perturbation Solution

*Proof.* Starting from the autonomous system Eq. 10 and assumption in 1, we have

$$\boldsymbol{\alpha} = \eta\left(-\log\boldsymbol{\alpha} - \boldsymbol{\alpha}\mathbf{A}\right) \tag{54}$$

where

$$\mathbf{A} = \lambda^{-1}\mathbf{U}(\mathbf{S}^2 + \lambda\mathbf{I})\mathbf{U}^T \tag{55}$$

Since the off-diagonal terms are of order $\delta$, we treat them as a perturbation. The unperturbed system has a solution $\boldsymbol{\alpha}_0$ given by case 2

$$\alpha_{0,i}^* = \frac{\mathcal{W}(A_{ii})}{A_{ii}}. \tag{56}$$

We can linearize the autonomous system Eq. 10 around the unperturbed solution to find,

$$\dot{\boldsymbol{\alpha}} \simeq \eta \left( -\log\left(\boldsymbol{\alpha}_0^*\right) - \boldsymbol{\alpha}_0^* \odot \operatorname{diag}\left(\boldsymbol{A}\right) + \frac{\mathrm{d}}{\mathrm{d}\boldsymbol{\alpha}} \left[ -\log\left(\boldsymbol{\alpha}\right) - \boldsymbol{\alpha} \odot \operatorname{diag}\left(\boldsymbol{A}\right) \right] \Big|_{\boldsymbol{\alpha}_0^*} \left(\boldsymbol{\alpha} - \boldsymbol{\alpha}_0^*\right) \right), \tag{57}$$

$$\dot{\boldsymbol{\alpha}} \simeq \eta \left( 1 - \log\left(\boldsymbol{\alpha}^*\right) - \left( \operatorname{diag}\left(\boldsymbol{A}\right) + \boldsymbol{\alpha}_0^{*-1} \right) \odot \boldsymbol{\alpha} \right). \tag{58}$$

We then apply the perturbation given by off-diagonal terms of $\mathbf{A}$ to obtain

$$\dot{\boldsymbol{\alpha}} \simeq \eta \left( 1 - \log\left(\boldsymbol{\alpha}^*\right) - \boldsymbol{\alpha} \left[ \mathbf{A} + \operatorname{diag}\left(\boldsymbol{\alpha}_0^{*-1}\right) \right] \right), \tag{59}$$

where $\operatorname{diag}\left(\boldsymbol{\alpha}_0^{*-1}\right)$ is the diagonal matrix obtained from $\boldsymbol{\alpha}_0^{*-1}$ and where the inverse is applied element by element.

Solving for $\dot{\boldsymbol{\alpha}} = 0$, we obtain the solution

$$\boldsymbol{\alpha}^* = \left(1 - \log\left(\boldsymbol{\alpha}_0^*\right)\right) \left[ \mathbf{A} + \operatorname{diag}\left(\boldsymbol{\alpha}_0^{*-1}\right) \right]^{-1}. \tag{60}$$

$\square$

## C.10  Thm. 2 Gradient Starvation Regime

**Theorem** (Gradient Starvation Regime). *Consider a neural network in the linear regime, trained under cross-entropy loss for a binary classification task. With definition 1, assuming coupling between features 1 and 2 as in Eq. 15 and $s_1^2 > s_2^2$, we have,*

$$\frac{\mathrm{d}z_2^*}{\mathrm{d}s_1^2} < 0, \tag{61}$$

*which implies GS.*

*Proof.* From lemma 1, and with $\mathbf{U}$ given by Eq. 15, we find that the perturbatory solution for the fixed point is

$$\alpha_1^* = \left[ \lambda \left( W \left( \frac{\lambda + \delta^2 \left( s_1^2 - s_2^2 \right) + s_2^2}{\lambda} \right) + 1 \right) \left( \delta \sqrt{1 - \delta^2} \left( s_2^2 - s_1^2 \right) + \lambda e^{W \left( \frac{\lambda + \delta^2 \left( s_1^2 - s_2^2 \right) + s_2^2}{\lambda} \right)} \right) \right.$$
$$\left. \left( W \left( \frac{\lambda + \delta^2 \left( -s_1^2 \right) + \delta^2 s_2^2 + s_1^2}{\lambda} \right) + 1 \right) \right) \right] \left[ \delta^2 \left( \delta^2 - 1 \right) \left( s_1^2 - s_2^2 \right)^2 + \right.$$
$$\left( \lambda + \delta^2 \left( s_1^2 - s_2^2 \right) + \lambda e^{W \left( \frac{\lambda + \delta^2 \left( s_1^2 - s_2^2 \right) + s_2^2}{\lambda} \right)} + s_2^2 \right)$$
$$\left. \left( \lambda + \delta^2 s_2^2 - \left( \delta^2 - 1 \right) s_1^2 + \lambda e^{W \left( \frac{\lambda + \delta^2 \left( -s_1^2 \right) + \delta^2 s_2^2 + s_1^2}{\lambda} \right)} \right) \right]^{-1}$$

$$\alpha_2^* = \left[ \lambda \left( W \left( \frac{\lambda + \delta^2 \left( -s_1^2 \right) + \delta^2 s_2^2 + s_1^2}{\lambda} \right) + 1 \right) \left( \delta \sqrt{1 - \delta^2} \left( s_2^2 - s_1^2 \right) + \lambda e^{W \left( \frac{\lambda + \delta^2 \left( -s_1^2 \right) + \delta^2 s_2^2 + s_1^2}{\lambda} \right)} \right) \right.$$
$$\left. \left( W \left( \frac{\lambda + \delta^2 \left( s_1^2 - s_2^2 \right) + s_2^2}{\lambda} \right) + 1 \right) \right) \right]$$
$$\left[ \delta^2 \left( \delta^2 - 1 \right) \left( s_1^2 - s_2^2 \right)^2 + \left( \lambda + \delta^2 \left( s_1^2 - s_2^2 \right) + \lambda e^{W \left( \frac{\lambda + \delta^2 \left( s_1^2 - s_2^2 \right) + s_2^2}{\lambda} \right)} + s_2^2 \right) \right.$$
$$\left. \left( \lambda + \delta^2 s_2^2 - \left( \delta^2 - 1 \right) s_1^2 + \lambda e^{W \left( \frac{\lambda + \delta^2 \left( -s_1^2 \right) + \delta^2 s_2^2 + s_1^2}{\lambda} \right)} \right) \right]^{-1}$$

We have found at Eq. 11 that the corresponding steady-state feature response is given by

$$\mathbf{z}^* = \frac{1}{\lambda}\mathbf{S}^2\mathbf{U}^T\boldsymbol{\alpha}^{*T} \tag{62}$$

In the perturbatory regime $\delta$ is taken to be a small parameter. We therefore perform a first-order Taylor series expansion of $\mathbf{z}^*$ around $\delta = 0$ to obtain

$$z_1^* = \frac{\delta s_1^2\left(W\left(\frac{\lambda+s_2^2}{\lambda}\right)+1\right)\left(\lambda+\lambda e^{W\left(\frac{\lambda+s_1^2}{\lambda}\right)}+s_2^2\right)}{\left(\lambda+\lambda e^{W\left(\frac{\lambda+s_1^2}{\lambda}\right)}+s_1^2\right)\left(\lambda+\lambda e^{W\left(\frac{\lambda+s_2^2}{\lambda}\right)}+s_2^2\right)} + \frac{\lambda s_1^2 e^{W\left(\frac{\lambda+s_2^2}{\lambda}\right)}\left(W\left(\frac{\lambda+s_1^2}{\lambda}\right)+1\right)\left(W\left(\frac{\lambda+s_2^2}{\lambda}\right)+1\right)}{\left(\lambda+\lambda e^{W\left(\frac{\lambda+s_1^2}{\lambda}\right)}+s_1^2\right)\left(\lambda+\lambda e^{W\left(\frac{\lambda+s_2^2}{\lambda}\right)}+s_2^2\right)} \tag{63}$$

$$z_2^* = \frac{\lambda s_2^2 e^{W\left(\frac{\lambda+s_1^2}{\lambda}\right)}\left(W\left(\frac{\lambda+s_1^2}{\lambda}\right)+1\right)\left(W\left(\frac{\lambda+s_2^2}{\lambda}\right)+1\right)}{\left(\lambda+\lambda e^{W\left(\frac{\lambda+s_1^2}{\lambda}\right)}+s_1^2\right)\left(\lambda+\lambda e^{W\left(\frac{\lambda+s_2^2}{\lambda}\right)}+s_2^2\right)} - \frac{\delta s_2^2\left(W\left(\frac{\lambda+s_1^2}{\lambda}\right)+1\right)\left(\lambda+\lambda e^{W\left(\frac{\lambda+s_2^2}{\lambda}\right)}+s_1^2\right)}{\left(\lambda+\lambda e^{W\left(\frac{\lambda+s_1^2}{\lambda}\right)}+s_1^2\right)\left(\lambda+\lambda e^{W\left(\frac{\lambda+s_2^2}{\lambda}\right)}+s_2^2\right)} \tag{64}$$

Taking the derivative of $z_2^*$ with respect to $s_1$, we find

$$\frac{\mathrm{d}z_2^*}{\mathrm{d}s_1^2} = -\frac{\delta\lambda s_2^2\left(e^{W\left(\frac{\lambda+s_1^2}{\lambda}\right)}-e^{W\left(\frac{\lambda+s_2^2}{\lambda}\right)}\right)\left(W\left(\frac{\lambda+s_1^2}{\lambda}\right)+1\right)}{\left(\lambda+\lambda e^{W\left(\frac{\lambda+s_1^2}{\lambda}\right)}+s_1^2\right)^2\left(\lambda+\lambda e^{W\left(\frac{\lambda+s_2^2}{\lambda}\right)}+s_2^2\right)} \tag{65}$$

Knowing that the exponential of the $W$ Lambert function is a strictly increasing function and that $s_1^2 > s_2^2$, we find

$$\frac{\mathrm{d}z_2^*}{\mathrm{d}s_1^2} < 0. \tag{66}$$

$\square$

## C.11    Eq. 18 Spectral Decoupling

SD replaces the general L2 weight decay term in Eq. 5 with an L2 penalty exclusively on the network's logits, yielding

$$\mathcal{L}(\boldsymbol{\theta}) = \mathbf{1}\cdot\log\left[1+\exp\left(-\mathbf{Y}\hat{\mathbf{y}}\right)\right] + \frac{\lambda}{2}\|\hat{\mathbf{y}}\|^2.$$

The loss can be written as,

$$\mathcal{L}(\boldsymbol{\theta}) = \mathbf{1}\cdot\log\left[1+\exp\left(-\mathbf{Y}\hat{\mathbf{y}}\right)\right] + \frac{\lambda}{2}\|\boldsymbol{\Phi}\boldsymbol{\theta}\|^2,$$

$$= \mathbf{1}\cdot\log\left[1+\exp\left(-\mathbf{Y}\hat{\mathbf{y}}\right)\right] + \frac{\lambda}{2}\left\|\boldsymbol{S}\boldsymbol{V}^T\boldsymbol{\theta}\right\|^2,$$

$$\geq \mathbf{1}\cdot\left[H(\boldsymbol{\alpha})-\boldsymbol{\alpha}\mathbf{Y}\hat{\mathbf{y}}\right] + \frac{\lambda}{2}\left\|\boldsymbol{S}\boldsymbol{V}^T\boldsymbol{\theta}\right\|^2.$$

Optimizing $\mathcal{L}(\boldsymbol{\theta})$ wrt to $\boldsymbol{\theta}$ results in the following optimum,

$$\boldsymbol{\theta}^{*T} = \frac{1}{\lambda}\boldsymbol{\alpha}\mathbf{Y}\boldsymbol{\Phi}_0\boldsymbol{V}\boldsymbol{S}^{-2}\boldsymbol{V}^T,$$

which by substitution into the loss function, the dynamics of gradient ascent leads to,

$$\dot{\boldsymbol{\alpha}} = \eta\left(\log\frac{1-\boldsymbol{\alpha}}{\boldsymbol{\alpha}} - \frac{1}{\lambda}\boldsymbol{\alpha}\mathbf{U}\mathbf{S}^2\mathbf{S}^{-2}\mathbf{U}^T\right) = \eta\left(\log\frac{1-\boldsymbol{\alpha}}{\boldsymbol{\alpha}} - \frac{1}{\lambda}\boldsymbol{\alpha}\right),$$

where $\log$ and division are taken element-wise on the coordinates of $\boldsymbol{\alpha}$ and hence dynamics of each $\alpha_i$ is independent of other $\alpha_{j\neq i}$.