# OpenReview forum: "Gradient Starvation: A Learning Proclivity in Neural Networks"
_NeurIPS.cc/2021/Conference — NeurIPS 2021 Poster_

### Official Review · Reviewer_TXkC · 2021-07-09

**Rating:** 3
**Confidence:** 4

**Summary:**

The authors formalize the notion of gradient starvation, discuss some mathematical aspects, and provide a potential remedy. They use mathematical and empirical means to make their point.

**Limitations And Societal Impact:**

No for the limitations (see above), but there is no negative societal impact expected.

**Main Review:**

The idea of formalizing gradient starvation seems interesting. However, the paper does not deliver on its promises. In the abstract, it is written "Based on our proposed formalism, we develop guarantees for a novel but simple regularization method aimed at decoupling feature learning dynamics, improving accuracy and robustness in cases hindered by gradient starvation." There are, unfortunately, no such guarantees (see Section 3.4).

They do have mathematical insights for ridge-regularized CE loss (Theorems 1 and 2), but it is not clear to me to what extend these insights are relevant; for example, Theorem 1 hinges on a Taylor approximation (without discussion of whether that is appropriate), Lemma 1 hinges on Assumption 1 (without discussion of whether that is appropriate), Lemma 1 talks about an approximation (without discussion of what that even means), ...

In my opinion, this paper lacks the necessary mathematical rigor.

Besides, I cannot find a discussion of the optimization properties of (17). (On a related note, I think 3(d) in the checklist is not adequately replied to.)

**Time Spent Reviewing:**

1

---

> ### Author Response · Authors · 2021-08-11
> **Clarifications**
>
> We thank the reviewer for their time.
>
> > The paper does not deliver on its promises. In the abstract, it is written "Based on our proposed formalism, we develop guarantees for a novel but simple regularization method aimed at decoupling feature learning dynamics, improving accuracy and robustness in cases hindered by gradient starvation." There are, unfortunately, no such guarantees (see Section 3.4).
>
> There seems to be some misunderstanding on the key aspects of this submission.
> Contrary to what the reviewer states, we **do** provide the guarantee that spectral decoupling results in zero starvation. Specifically, Eq. 18 presents the dual dynamics at the presence of the proposed spectral decoupling. As pointed out immediately at line 198, $\alpha_i$’s are decoupled and, in fact, have closed-form solutions. Now, with the definition of gradient starvation, as presented in def 2, we note that $\frac{\partial z_j^*}{s_i} = 0$, and hence it is guaranteed that no starvation occurs.
>
>
> > Theorem 1 hinges on a Taylor approximation (without discussion of whether that is appropriate),
>
> In lines 145-148, we clearly outline that a Taylor approximation is justified since “fixed point is drawn closer towards the origin”. The non-linear terms in the vicinity of the origin are extremely small and it is typical to approximate Eq. 9 with Eq. 10. See e.g., the approximation at the bottom of page 3 of [21].
>
> > Lemma 1 hinges on Assumption 1
>
> In lines 155-158, we clearly outline two cases in which features’ learning does not interfere. As stated in lines 164-165, Assumption 1 is a perturbation of case (2), a natural extension to the case where learning one feature interferes with another.
>
> > Lemma 1 talks about an approximation (without discussion of what that even means)
>
> In lines 929-932 of the appendix, we discuss that we approximate around the *unperturbed solution* with a small amount of perturbation $\delta$ and hence a Taylor approximation is a natural choice when studying the properties of the *perturbed solution*.
>
> > Besides, I cannot find a discussion of the optimization properties of (17). (On a related note, I think 3(d) in the checklist is not adequately replied to.)
>
> The reviewer might have missed the reference we have provided for question 3 (d). We politely point out to the checklist where it reads, “The details of our experiments are included in the supplementary materials B.” Appendix B.5 clearly states that for the experiments and hyper-parameter search, an approximate number of 800 GPU-hours has been used. GPUs used for the experiments are NVIDIA-V100, mostly on internal clusters and partly on public cloud clusters. Also, in lines 823-824, 826, 835, and 842-844, we explain the used libraries and the hyperparameter search scheme and results.
>
> ---
>
> Once again, we appreciate your time. We want to make sure that your assessment of this submission is based on a clear understanding. Please let us know if any further clarification is needed.

---

> > ### Author Response · Authors · 2021-08-19
> > **Follow up**
> >
> > Dear reviewer,
> >
> > We would like to thank you once again for your time and reiterate that we are keen to hear your thoughts after reading our responses. We will, of course, remain at your disposal should you have any further questions and continue to provide clarifications as promptly as possible.

---

### Official Review · Reviewer_ZevV · 2021-07-19

**Rating:** 7
**Confidence:** 3

**Summary:**

This paper studies the phenomenon of gradient starvation in the context of training neural networks via gradient descent. The concept is that some features are learned first and help the model fit the data well enough that other features are “starved” of gradient and never get learned. While this concept existed previously, the authors do a good job of exploring it. They develop an interesting and simple regularization scheme to mitigate gradient starvation, essentially discouraging the model from being too confident in its predictions, and test it out in a number of settings from adversarial robustness to domain generalization.

**Limitations And Societal Impact:**

This seems like a paper that could benefit from a societal impacts section, as it deals directly with what types of features are learned or ignored when training neural networks. I also think more discussion of the extent to which gradient starvation is bad could be included.

**Main Review:**

Figure 1 provides a nice illustration of a case where gradient starvation should hurt generalization, especially in the ood setting where the half moons may move across the decision boundary. This perspective has sparked a lot of ideas in this reviewer about neural networks and generalization.

I would like to hear more of the authors thoughts on the relationship between gradient starvation and generalizable features. While the colored mnist experiment was tailor made such that the fast-to-learn features are the ones that don’t generalize, one could certainly concoct the opposite scenario where the first learned features are the good ones and the later learned features are overfitting. Indeed, this might even be expected as the authors point out. Being able to dial up or down how much gradient starvation is present seems like a valuable tool to me, but it’s not obvious that one should always try to minimize it as much as possible. Is there an argument for when gradient starvation is bad v. when it actually helps?

Could the authors comment on the relationship of their method to arxiv:1701.06548 which also penalizes confident predictions?



**Time Spent Reviewing:**

2

---

> ### Author Response · Authors · 2021-08-11
> **Clarification**
>
> We are thrilled to hear that you found this perspective interesting!
> In fact, several ongoing studies by independent researchers are building upon this work where due to anonymity, we cannot provide any references. We believe acceptance in this review cycle would benefit the NeurIPS community. Below, we do our best to address your questions, but before that, we encourage you to read the “to all the reviewers'' section first.
>
> > I would like to hear more of the authors thoughts on the relationship between gradient starvation and generalizable features. While the colored mnist experiment was tailor made such that the fast-to-learn features are the ones that don’t generalize, one could certainly concoct the opposite scenario where the first learned features are the good ones and the later learned features are overfitting. Indeed, this might even be expected as the authors point out. Being able to dial up or down how much gradient starvation is present seems like a valuable tool to me, but it’s not obvious that one should always try to minimize it as much as possible. Is there an argument for when gradient starvation is bad v. when it actually helps?
>
> This is an excellent intuition at the center of our message! We in fact discuss both beneficial and adverse consequences of gradient starvation in appendix A under the paragraph “Gradient Starvation: friend or foe?”. As you correctly pointed out, one may redefine the Colored MNIST task as a “color classification” task instead of “digit classification”. In that case, learning the color is the right thing to do, and gradient starvation is indeed beneficial.\
> Generally, gradient starvation can be viewed as an implicit regularizer if the learned features are sufficient to generalize to the test data. Otherwise, gradient starvation could have an unfavorable effect, which we observe empirically when some predictive features fail to be learned. A better understanding and control of gradient starvation and its impact on generalization offers promising avenues to address this issue with minimal assumptions. Indeed, our spectral decoupling method requires an assumption about feature imbalance but not to pinpoint them exactly, relying on modulated learning dynamics to achieve balance.
>
> > Could the authors comment on the relationship of their method to arxiv:1701.06548 which also penalizes confident predictions?
>
> Thank you for pointing out this work. It is closely related to spectral decoupling and we make sure to include that in the related work section. While entropy minimization does not have the guarantees that we provide for spectral decoupling, the mechanism hinges on the same concept. As discussed in the “response to all” section, intuitively, there are two ways for a network to reduce its loss: 1) Get more confident on correctly classified examples, or 2) Learn to classify new examples. Spectral decoupling prevents (1) and forces the network to learn new features to classify new examples correctly.
>
> > This seems like a paper that could benefit from a societal impacts section, as it deals directly with what types of features are learned or ignored when training neural networks. I also think more discussion of the extent to which gradient starvation is bad could be included.
>
> Thank you for this important remark. We in fact, discuss the societal impacts of gradient starvation in Appendix A, lines 789-797. In short, we believe that gradient starvation could have adverse effects the most when data is biased which can be mitigated by spectral decoupling.\
> Gradient starvation has immediate societal consequences as 1) neural networks are being used in critical applications, and 2) reliance on spurious features may go unrecognized. As a recent example, we would like to point you to an article by researchers at Cambridge [22] where they study more than 300 papers on “detecting whether a patient has COVID or not given their CT Scans”. According to the article, none of the papers were able to generalize from one hospital data to another since the models learn to latch on hospital-specific features. Hence, understanding the mechanisms behind the learning dynamics seems an essential step towards uncovering the internals of neural networks’ learning.

---

### Official Review · Reviewer_7wTW · 2021-07-22

**Rating:** 7
**Confidence:** 2

**Summary:**

This paper analyzes a phenomenon in classification with neural networks, when cross-entropy loss can be minimized by learning only some features correlated with the label (while other features correlated with the label also exist). In a sense, the NN is able to minimize the CE loss by confidently predicting the correct answer even when the margin is small. The paper proposes a method (spectral decoupling) to address this,  which penalizes the network's logits instead of the network's weights, which leads it to improved out of distribution performance e.g. adversarial robustness, and test performance on MNIST when the train and test environments have different color biases (correlations or reverse correlations of color with label).

**Limitations And Societal Impact:**

Yes

**Main Review:**

This is a mostly nice paper with a few things that could be improved.

--
Good:
- The example in Figure 1 is great. It makes the effects of the problem very clear from the get go. It's great that this figure is generated with real data from training runs.
- The mathematical analysis clearly highlights the problem should exist
- Spectral decoupling is a very natural idea / regularization penalty
- SD appears to perform very well across a variety of well-chosen tasks that measure out-of-distribution performance
- The discussion "on gradient starvation and no free lunch theorem" is maybe a bit unusual for a NeurIPS paper but is, I believe, profound, especially given the various methods like IRM that are being trialled to learn rules which generalize out of distribution or to new environments. Sometimes we don't need a distribution over environments (which let the learner deduce what signals/correlations do not generalize), we just need to learn to be able to use *all* the signals/correlations that are present in one training dataset.

--
Feedback and questions:
- In first para, the sentence about personal experiences seems out of place. Delete or fold into previous sentence, discussing phenomena from cited papers (or blog posts or other sources) about NNs learning to classify from spurious information.
-  Don't capitalize Dynamical Systems, only capitalize proper nouns.
- In the abstract, it sounds like you discovered / defined gradient starvation. In the second para, it sounds like you are just extending gradient starvation / the analysis of feature imbalance from regression to cross entropy loss. It would be good to try to pick one perspective and make sure it aligns between abstract and intro.
- 62: "We argue that this occurs because cross-entropy loss leads to gradients “starved” of information.." This paragraph could do with referencing later in the text where this argument occurs (section 3). It's also worth noting that while you do give evidence in S3 that GS should be a problem, you don't give any evidence that GS fully accounts for the non-curved-boundary problem - could consider rewriting as "we argue that one reason this occurs" or similar.
- Could you provide a bit more intuition for the effects of definition 2? Feature i and feature j can conflict (in the sense that increase in strength of i has detrimental effect on j) without necessarily preventing j from being learned. I don't see an obvious reason why definition 2 should necessarily prevent a NN from learning all meaningful features and reaching a robust solution. When does definition 2 lead to some features not being learned, and lead to lack of robustness? There are some missing steps here.
- It would be good to provide some experimental measurement / analysis of gradient starvation. What does the quantity in definition 2 look like for NNs trained on some datasets with and without SD?
- In the analysis in 3.2 and 3.3 there are a bunch of assumptions, and it would be useful to have a bit more text explaining e.g. exactly how the rationale for GS with two slightly-coupled features extends to higher dimensions.
- Are there any simpler explanations for why spectral decoupling (pushing down on the logits) works and e.g. achieves a curved decision boundary in the toy example?
- You should mention REx in the text around 242, as this performs competitively to SD - does REx have any constraints beyond SD?
- In CelebA, you use a variant of SD where a target logit and regularizer weight is tuned independently for every class. Can you provide intuition for why this performs better for this task? What were the optimal hyperparameters? Can you compare to the original SD? it would be interesting to know whether vanilla SD does much for this task or not. Separately tuning two extra hyperparameters per class is a massive disadvantage for real problems, even if the number of classes are small, as it will multiply the number of runs required for a good hyperparameter search.

--
I am not very up to date / well versed in the feature imbalance / out-of-distribution generalization literature, such as IRM etc, and can't comment much on originality beyond what the authors' themselves have written, so my confidence is low. However it seems to me like this paper provides some interesting analysis of training dynamics under CE loss, and motivates a simple regularizer which performs very well in practice. This could easily become a standard technique for training classifiers, and could be of very high impact. Therefore I recommend acceptance.

**Time Spent Reviewing:**

5

---

> ### Author Response · Authors · 2021-08-11
> **Clarifications**
>
> We are delighted to read your review and have found your comments very constructive! Below, we do our best to address your questions, but before that, we encourage you to read the “to all the reviewers'' section first.
>
> > In first para, the sentence about personal experiences seems out of place. Delete or fold into previous sentence, discussing phenomena from cited papers (or blog posts or other sources) about NNs learning to classify from spurious information. Don't capitalize Dynamical Systems, only capitalize proper nouns.
>
> Thanks for your suggestions. We will make sure to provide proper citations when motivating the problem. We will also verify that the notation and names remain consistent throughout the text.
>
> > In the abstract, it sounds like you discovered / defined gradient starvation. In the second para, it sounds like you are just extending gradient starvation / the analysis of feature imbalance from regression to cross entropy loss. It would be good to try to pick one perspective and make sure it aligns between abstract and intro.
>
> In the revised version, we will clarify that this is the cross-entropy loss that suffers from feature imbalance. When using the squared-error loss, different features with different strengths are still learned at different rates. However, unlike cross-entropy, their learning does not interfere, i.e., the dynamics of a feature is only a function of that feature strength alone.
>
>
>
> > 62: "We argue that this occurs because cross-entropy loss leads to gradients “starved” of information.." This paragraph could do with referencing later in the text where this argument occurs (section 3). It's also worth noting that while you do give evidence in S3 that GS should be a problem, you don't give any evidence that GS fully accounts for the non-curved-boundary problem - could consider rewriting as "we argue that one reason this occurs" or similar.
>
> We acknowledge that it might not be clear for the reader from the onset. We will be more careful in presenting the idea / motivation.
>
> > Could you provide a bit more intuition for the effects of definition 2? Feature i and feature j can conflict (in the sense that increase in strength of i has detrimental effect on j) without necessarily preventing j from being learned. I don't see an obvious reason why definition 2 should necessarily prevent a NN from learning all meaningful features and reaching a robust solution. When does definition 2 lead to some features not being learned, and lead to lack of robustness? There are some missing steps here.’
>
> We acknowledge that the results provided in Thm 2 are in fact local and show that a slight increase in strength of a feature (s_i) results in a decrease in the learning of another coupled feature (z_j). However, we note that rates of learning are exponential functions of s_i. Intuitively, if a training example has two features with strength s_i > s_j, the feature z_i is learned much faster, resulting in almost zero gradient for this training example, possibly even before the learning of z_j is started.
>
> > It would be good to provide some experimental measurement / analysis of gradient starvation. What does the quantity in definition 2 look like for NNs trained on some datasets with and without SD?
>
> Thank you for pushing this question which led to a novel experimental validation. We have included a new plot in the anonymous repo of our submission [link](https://github.com/gradientstarvation/GS/blob/master/rebuttal/spectrum.pdf). The idea of the experiment is to monitor the rank (singular values) of the NTK feature matrix over the course of training. For the task of Colored MNIST, when trained without spectral decoupling, a single feature, which most probably corresponds to the color, takes over the dynamics resulting in a low-rank feature matrix. That single feature dominates the prediction of the network. On the other hand, the spectral decoupling restrains the dominant feature and allows the feature matrix to have a higher rank.
>
> > In the analysis in 3.2 and 3.3 there are a bunch of assumptions, and it would be useful to have a bit more text explaining e.g. exactly how the rationale for GS with two slightly-coupled features extends to higher dimensions.
>
> We will revise the manuscript to include some explanations on how the analysis could extend to higher dimensions. Note that we touch on that point briefly at the end of section 3.3 (lines 178-181). In particular, the theory is also valid in the case where pairwise interactions relate more than two features. In this case, the matrix U can be decomposed in blocks, each of a similar form to the one we considered. The block can be considered separately, and the theorem applies. We leave it open for future work to extend the theory to a less restrictive setting.
>
> > Are there any simpler explanations for why spectral decoupling (pushing down on the logits) works and e.g. achieves a curved decision boundary in the toy example?
>
> In the section "to all reviewers", we discuss this point in detail. We make sure to articulate this intuition in the main text properly.
>
> > You should mention REx in the text around 242, as this performs competitively to SD - does REx have any constraints beyond SD?
>
> We mistakenly omitted the discussion on REx during editing but will include it in the final version. In short, REx is a training method that encourages learning “invariant” representations. It is very similar to IRM and requires access to multiple training environments in order to quantify the concept of “invariance”. While REx scores are close to the ones obtained with our spectral decoupling (SD), we point out that SD does not require explicitly distinct training environments and thus is a more flexible method.
>
> > In CelebA, you use a variant of SD where a target logit and regularizer weight is tuned independently for every class. Can you provide intuition for why this performs better for this task? What were the optimal hyperparameters? Can you compare to the original SD? it would be interesting to know whether vanilla SD does much for this task or not. Separately tuning two extra hyperparameters per class is a massive disadvantage for real problems, even if the number of classes are small, as it will multiply the number of runs required for a good hyperparameter search.
>
> During the experiments, we found that for the CelebA dataset, classes are imbalanced: 10875 examples for class 0 and 1925 examples for class 1; meaning a ratio of 5.65.
> That is why we decided to penalize examples of each class separately with different coefficients. We also found that penalizing the outputs' distance to different values $\gamma_0$ and $\gamma_1$ helps the generalization. As stated in lines 842-844, the hyperparameter search has resulted in the following values:  2.5  and 0.44. Interestingly, thanks to your question, we realized that the ratio of $\gamma_1$ to $\gamma_0$ is 5.68 which is very close to the ratio of the classes. This hints towards the idea that one might be able to reduce the number of hyperparameters by half. We will provide a discussion on this in the appendix.
>
> ---
>
> Once again, thank you for your fair assessment of this work. We note that despite your detailed review, your confidence score is low. Here, we volunteer to provide some context about this work; The results presented here have been highly well-received during workshop presentations. Several ongoing studies by independent researchers are building upon this work where due to anonymity, we cannot provide any references. Hence, we believe the NeurIPS community could benefit from this work. We look forward to any further discussions that would help your evaluation.

---

### Official Review · Reviewer_i1Jd · 2021-08-01

**Rating:** 5
**Confidence:** 3

**Summary:**

The authors identify a phenomenon that happens during training with GD, which they call gradient starvation (GS). This phenomenon happens when training captures only a subset of the features, while other features which may be beneficial for prediction are not captured. There is a theoretical explanation of the phenomenon for the simple setting of two features. A regularization method called spectral decoupling (SD) is introduced to help mitigate GS. Several experiments are shown on a couple of datasets that SD improves performance in several different settings.


**Limitations And Societal Impact:**

The authors adequately addressed the limitations and potential negative societal impact of their work.

**Main Review:**


Pros:

1) The paper brings forward an interesting phenomenon that may happen during training with gradient methods. As far as I know, this is a novel phenomenon and is well defined for the NTK regime.
2) The example of the two “moon” shapes illustrates the phenomenon very well. The analysis in Section 3 seems to explain why this phenomenon happens for the two feature case (which is the case here). Also, the SD solution indeed helps in mitigating the problem for this case.
3) The experiments are thorough, and many datasets are being used including CIFAR-10/100, CelebA, and colored MNIST.

Cons

1) The definition itself of GS, which is the main definition of the paper, seems to be focused only on the NTK regime, where all the weights are fixed except for the last layer. In Definition 1, the “features” and “response to a feature” will probably change during real training, and this is not captured in the definition. This goes on to Definition 2, as it seems that GS may only happen during initialization, and it is not clear whether this phenomenon still happens during training of real networks in the “rich” regime (which is more realistic than the NTK) where all the weights change.

2) The analysis in Section 3 seems to be focused only on the NTK regime. This is OK if technically this is what the authors can show, but they should at least explain, even intuitively, what happens during training where all the weights change. Otherwise, it is not clear how the GS phenomenon behaves in the non-NTK regime.

3) Theorem 2 is very limited to the case with two features, and a very specific matrix U. I don’t think this gives enough motivation as to when the GS phenomenon happens. Is it possible to extend this result to a general number of features where the matrix U is “almost” orthogonal? Or maybe to other more general cases?

4) It also seems that the intuition for spectral decoupling is in the NTK regime, i.e. the optimal \theta^* is given in terms oh \Phi_0 which are the weights at initialization. These may not be the optimal weights in the non-NTK regime.

5) My main problem with the experimental part is the methods to which SD is being compared. SD is a simple regularization technique, so I would expect to first compare it to other regularization techniques, before comparing it to other methods. Such techniques include, for example, l2/l1 regularization, batch-norm, and dropout. This is done for the two “moon” shaped clusters which is a synthetic dataset specifically aimed at presenting the GS phenomenon. However, I am not convinced that these simple regularization methods wouldn’t outperform SD.

There are also a couple of issues with the theoretical part and the presentation of the paper:
1) What is m in Eq (2)? Is it the number of parameters? Because the previous sentence presents the NTK as the limit of the infinite width. If this is not the limit, but a finite number of parameters are considered, I think the authors should state how big this number should be w.r.t n.

2) What is c in Eq (8)? It does not seem to be defined in the main text.

3) In Theorem 1, an “attractive fixed point” is not defined. It seems (from the appendix) that it is a fixed point with negative definite Jacobian, this should be defined properly in the main text.

4) Also regarding Theorem 1, why is it important? Is it used somewhere in the paper? The context of this theorem is not clear.

5) The derivations of Eq (6)-(10) in the appendix are not very clear and confusing. It would be much better to phrase these as a theorem/lemma and prove them rigorously, stating exactly what is being assumed.

6) Lemma 1 and Theorem 2 are written imprecisely w.r.t to the perturbations. How small \delta should be? Does its magnitude depend on some parameters of the problem, and what is the dependence?

7) Table 2 - What is REx? It would be better to write a short sentence in the main text explaining it.

8) A notation section would be really helpful in this paper to prevent ambiguity. Some notations which are used but not defined: diag (y), (V)_j. , (U)_.j , \odot, Vec(W)

To conclude, I think this paper presents an interesting phenomenon and demonstrates it both theoretically and empirically. On the other hand, the theoretical analysis is very limited, and in the experimental part, some basic comparisons are missing. To improve this paper I suggest to (1) Analyze either the non-NTK case or a more general case than presented in Theorem 2. (2) Give a better motivation for the SD solution, currently, the motivation is given only for the NTK regime (while in practice not only the last layer is being trained). (3) Conduct more experiments that demonstrate why the SD regularization technique outperforms other simple regularization techniques on real datasets.

--------------edit after rebuttal----------

I read the authors' response and the other reviews. I do appreciate the effort that the authors made with their response. Albeit, my concerns about the limitations of the NTK regime, and of the theoretical part remain. Hence, I decided to retain my score.


**Time Spent Reviewing:**

5

---

> ### Author Response · Authors · 2021-08-11
> **Clarifications**
>
> We are happy that you have found our results interesting. Below, we do our best to address your questions adequately. We encourage you to read the “To all reviewers” section first and return here for further discussion.
>
> > The definition itself of GS, seems to be focused only on the NTK regime, where all the weights are fixed except for the last layer. […] as it seems that GS may only happen during initialization [...]
>
> We like to clarify that the NTK regime is different from fixing all the layers except the last layer. Instead, NTK refers to a regime where all parameters **do change**, but for wide enough networks, the tangent feature matrix ($\Phi_0$ in Eq. 3) remains within a bounded neighborhood throughout the training. Thus, we only assume that $\Phi_0$ is fixed, but all parameters (the vector $\theta$) change.
> In Def 2, on the contrary, we show that the final response at the convergence point ($\theta^*$) suffers from Gradient Starvation. We make sure to make this point crystal clear in the final version.
>
> > The analysis in Section 3 seems to be focused only on the NTK regime. [...] they should at least explain, even intuitively, what happens during training where all the weights change.
>
> We highlight that all the experiments of this manuscript are done in the “rich” regime, validating the theoretical results whose derivations are only possible thanks to the NTK theory and its associated assumptions.
>
> > Theorem 2 is very limited to the case with two features, and a very specific matrix U. [...] Is it possible to extend this result to a general number of features where the matrix U is “almost” orthogonal?
>
> We thank the reviewer for these insightful remarks and the constructive propositions. About the structure of the matrix U: we should highlight that U is the result of a singular value decomposition (SVD) on the tangent features matrix and hence is orthogonal by definition. Therefore, U's orthogonality is not an assumption of orthogonality of direct perturbations but instead is a representation of **any** perturbation on an appropriate orthogonalized basis.\
> That being said, as explained in Assumption 1, we consider a class of perturbations that are based on permutation matrices and perturbations of these. A natural extension of our theory would be to generalize to a larger set of perturbations, as suggested by the reviewer. We will introduce a specific discussion on this point.\
> On extension to more than two features, we leave it open for this to be carried out in future work. However, as briefly stated in lines 172-173, the results can be readily extended to two blocks of features interfering with each other.
> While it is true that theoretical results are limited to simpler cases, we show that it holds experimentally, even when the matrix U strongly deviates from identity, see Appendix B1.
>
> > It also seems that the intuition for spectral decoupling is in the NTK regime [...]
>
> We agree and acknowledge that the NTK regime imposes limitations. However, we would like to draw your attention to a few points we believe motivate this approach:
> - First, we note that in the limit of “large width,” NTK states that $\Phi_0$ remains unchanged during training, but $\theta$ still changes.
> - The assumption of “large width”, in fact, holds in many practical applications as in most cases, neural networks are highly overparameterized (see, e.g. [14, 15]).
> - We test spectral decoupling in standard training regimes in all our experiments and demonstrate its intended effect and advantages. In fact, for Colored-MNIST, Cifar experiments, and CelebA experiments, the architecture, including the width and the training details, are exactly those presented in their original papers.
> -Finally, we note that neural networks in non-NTK regime could exhibit complex inner dynamics which lies outside of the focus of our work. In our work, the focus is more on the loss function itself, showing the deficiencies of the cross-entropy.
>
> Hence, while we share your concerns, we strongly believe that using NTK to derive theoretical results that are empirically verified in rich regimes is a valuable contribution and opens the door for exciting future work.
>
> > My main problem with the experimental part is the methods to which SD is being compared. [...]
>
> We apologize for any confusion about the context of the experiments. It is known that the problems of Colored-MNIST, CelebA, and Cifar robustness are difficult and common regularization methods fail to improve their performance. \
> For Colored-MNIST, see Tab. 1 of [16] and Tab. 4 of [17].\
> For CelebA, see Tab. 1 of [18], column: ERM, rows:standard and L2 regularization.\
> SD, despite its simplicity, performs outstandingly better and must be compared in a different league than the common L1/L2 regularizations. That is why we compare SD with more sophisticated training algorithms such as IRM and Group DRO.
>
> > What is m in Eq (2)? [...]
>
> We make sure to be more precise in our definition of the feature matrix and NTK. More accurately, the changes in the kernel is inversely proportional to the square root of the width: See section 8.2 of [19] on “Building up the NTK from finite width.”
>
> > What is c in Eq (8)? [...]
>
> We apologize for the mistake. It should be $\alpha$.
>
> > In Theorem 1, an “attractive fixed point” is not defined. [...]
>
> In the context of dynamical systems, “attractive fixed point” is rather well-known. As you correctly pointed out, it refers to fixed points of a system at which the real part of the eigenvalues are negative, meaning that it attracts the dynamics along all the directions [20]. We make sure to provide appropriate pointers for the NeurIPS audience.
>
> > Also regarding Theorem 1, why is it important? [...]
>
> This is a good point that we will clarify. The analysis following Theorem 1 focuses on fixed points, irrespective of their attractive or repulsive character. Theorem 1 guarantees that these fixed points are attractors, which amounts to them being minimas and not maximas of the loss function.
>
> > The derivations of Eq (6)-(10) in the appendix are not very clear and confusing. [...]
>
> Thank you for pointing this out. We will be presenting Eq. 8 as a lemma as it is an important intermediary result. As for Eq. 9 and 10, we will make the assumptions behind their derivation as clear as possible and will provide their implications right after they are stated. In addition, we note that these derivations follow established methods of dual optimization presented in [12, 1].
>
> > Lemma 1 and Theorem 2 are written imprecisely w.r.t to the perturbations. [...]
>
> We will make important efforts to strengthen the argument following this comment. Concretely, in this case, the assumption is that $0 < \delta << 1$ irrespective of the other parameters of the problem. The validity of this approximation follows from the implicit function theorem, an argument often appealed to in perturbative approaches. Given the existence of a solution $\alpha_0$ as $\delta = 0$, there is a neighborhood around $\delta = 0$ for which $\alpha(\delta)$ is well defined. We assume that the dynamics are in this neighborhood.
>
> > Table 2 - What is REx? [...]
>
> We thank the reviewer for pointing out this shortcoming, we mistakenly omitted the discussion on REx during editing but will include it in the final version. In short, REx is a training method that encourages learning “invariant” representations. It is very similar to IRM and requires access to multiple training environments to quantify the concept of “invariance”. While REx scores are close to the ones obtained with our spectral decoupling (SD), we point out that SD does not require explicitly distinct training environments and thus is a more flexible method.
>
> > A notation section would be really helpful in this paper to prevent ambiguity. [...]
>
> We make sure to provide a notation section in the final version.
>
> > To improve this paper I suggest to (1) Analyze either the non-NTK case or a more general case than presented in Theorem 2.
> (2) Give a better motivation for the SD solution, currently, the motivation is given only for the NTK regime (while in practice not only the last layer is being trained).
> Conduct more experiments that demonstrate why the SD regularization technique outperforms other simple regularization techniques on real datasets.
>
> We point out an important experiment conducted in lines 249-264 where we perform an ablation study to understand how SD learns the core features in addition to the spurious features. In short, in the Colored MNIST task, SD prevents the network from getting overconfident on the spurious feature (color) and encourages the network to learn new features (generalizable shape features). The reason is that learning shape features would decrease the overall loss but does not result in over-confident predictions.
>
> In response to the perceived limited nature of our analysis, we trust that the detailed replies provided to the technical points made above will shed light on the scope and applicability of our derivations. While the suggestions made to improve the assumption sets are desirable, we respectfully point out the difficulty they represent. In the end, we strongly believe that contributions are significant despite the limitations of NTK, that empirical experiments strongly support them, and that they provide concrete next steps for future work. Finally, we thank the reviewer for their comments that will help us shape discussion points, and we hope they will agree with the value of our current contribution in light of our responses.

---

> > ### Author Response · Authors · 2021-08-19
> > **Follow up**
> >
> > Dear reviewer,
> >
> > We would like to thank you once again for your time and reiterate that we are keen to hear your thoughts after reading our responses. We will, of course, remain at your disposal should you have any further questions and continue to provide clarifications as promptly as possible.

---

### Official Review · Reviewer_PD11 · 2021-08-01

**Rating:** 4
**Confidence:** 3

**Summary:**

The paper investigates the problem of "gradient starvation" while training neural networks. This issue arises when only a subset of the input features comes into play in the final classification boundary of the network. The authors argue that this phenomenon is connected to properties of the eigenvalues of the NTK Gram matrix, and they investigate expressions for training dynamics in the case of a regularized cross-entropy loss function. They propose a regularization strategy that is meant to overcome the effects of gradient starvation, and they present experimental results on several datasets.

**Limitations And Societal Impact:**

Yes

**Main Review:**

The paper tackles an interesting and important topic that also has practical relevance. It is generally clear and well written, but I have concerns regarding its technical contributions. Here are some questions that I hope the authors can address:

* Theoretical development: although the authors claim to study deep neural networks, they immediately consider an NTK regime in which the network is a linear function of the parameters. As such, the analysis is essentially only presented for a linear model (with kernel features).
* The analysis of the training dynamics uses a variational characterization of the loss. It is not clear to me whether the dynamics in the parameter $\alpha$ are in any way related to the dynamics for the original parameters.
* Regarding the dynamics, isn't it the case that the loss in eq. 5 (strongly) convex in the parameters $\theta$? If so, there should be a single critical point corresponding to a global minimum and the dynamics should be completely irrelevant (maybe I'm misunderstanding here).
* Theorem 2: is this result stating that GS always occurs?
* What is the difference (if any) between using cross-entropy loss vs. a square loss for regression? The authors claim that it is important to focus on cross-entropy, but it is not clear from their analysis whether there is any significant difference from the perspective of GS.
* The proposed "spectral decoupling" regularization seems to be a simple L2 regularization in output space rather than in parameter space.

Other questions:
* l.74 "Excessive invariance": in a classification setting, I believe it will almost always be the case that points outside the data distribution will be classified with high confidence, simply by virtue of the fact that there is a full-dimensional region of inputs for each label, whereas the data is typically lower dimensional.
* Experimental results: does "vanilla cross-entropy" include the weight-decay regularization?
* l.235: "Correlated with .9 probability"? What does this mean?
* In Table 2, why is REx not discussed?

**Time Spent Reviewing:**

2

---

> ### Author Response · Authors · 2021-08-11
> **Clarifications**
>
>
> We are happy that you have found our results interesting. Below, we do our best to address your questions adequately. We encourage you to read the “To all reviewers” section first and return here for further discussion.
>
>
> > Theoretical development: although the authors claim to study deep neural networks, they immediately consider an NTK regime in which the network is a linear function of the parameters. As such, the analysis is essentially only presented for a linear model (with kernel features).
>
> We thank the reviewer for this remark and acknowledge that the NTK regime is equivalent to studying a linearized model. Nevertheless, NTK has proven to be a powerful tool used in several contexts where mechanisms are first identified analytically in this limited regime and then verified to hold in more general cases empirically ([9, 10, 11], to name a few). We employ the same approach here: first developing a tractable theory for gradient starvation and spectral decoupling in approximate regimes and validating in detailed experiments. Importantly, as the reviewer points out, we acknowledge the limitations of NTK. We will incorporate a clear discussion on these limitations in the revised manuscript. Despite these limitations, we strongly believe that the identified mechanisms have real-world impacts, are novel, and benefit the NeurIPS community.
>
>
>
> > The analysis of the training dynamics uses a variational characterization of the loss. It is not clear to me whether the dynamics in the parameter α are in any way related to the dynamics for the original parameters.
>
>
> As shown in line 138, the solution gradient descent converges to is a **weighted** sum of the training examples ($\theta^* = \sum \alpha_i x_i$). The dual approach suggests optimizing $\alpha$s (example contributions) directly instead of the $\theta$.
> As shown in Eq. 8 (Lemma 3 of [12]), the solution of the dual and primal dynamics are guaranteed to match. However, the intermediate dynamics might be and probably are different.
> Nevertheless, that is not a concern because Thm 2 studies the converged solution and not the intermediate dynamics.
>
>
> > Regarding the dynamics, isn't it the case that the loss in eq. 5 (strongly) convex in the parameters θ? If so, there should be a single critical point corresponding to a global minimum and the dynamics should be completely irrelevant (maybe I'm misunderstanding here).
>
> This is an important point that you mentioned. As discussed above, although the cross-entropy loss is convex, it does not admit an analytical solution, even in a simple logistic regression [2]. Importantly, it also does **not** have a finite solution when the data is linearly separable [3] (which is the case in high dimensions [4]). As such, our study is concerned with characterizing the solutions that the training algorithm converges to. A dual optimization approach enables us to describe these solutions in terms of the contribution of the training examples [5]. While primal and dual dynamics are not guaranteed to match, the solution they converge to is guaranteed to match [6], which is what our theory builds upon.
>
>
> > Theorem 2: is this result stating that GS always occurs?
>
> Thm 2 states that gradient starvation happens under the assumptions that “there is coupling between features 1 and 2” and “one is stronger than the other one”. Eq. 66 in the appendix suggests that the more coupling ($\delta$) we have, the more starvation occurs. However, as discussed in lines 781-788, that does not necessarily hurt the learning and generalization.
>
> > What is the difference (if any) between using cross-entropy loss vs. a square loss for regression? The authors claim that it is important to focus on cross-entropy, but it is not clear from their analysis whether there is any significant difference from the perspective of GS.
>
> This is an important point that we make sure to articulate clearly in the main text. As explained in “to all reviewers” section, gradient starvation does not happen for squared-error loss.
>
> > The proposed "spectral decoupling" regularization seems to be a simple L2 regularization in output space rather than in parameter space.
>
> Exactly! This is what makes the implementation of spectral decoupling so appealing for practitioners. Importantly, our novel contribution is to specifically identify why this regularization scheme is so effective. The gradient starvation derivations are novel, they identify an established problem in many applications, and the proposed spectral decoupling is a direct consequence of this analysis. Therefore, we believe that the simplicity of the proposed spectral decoupling regularization is an elegant manifestation of previously perplexing learning mechanisms under cross-entropy loss.
> Other independent studies, that we cannot cite to preserve the anonymity, have studied several regularization methods including spectral decoupling and have arrived at the conclusion that our proposed spectral decoupling method should become a standard technique for medial applications in which reliance on spurious features is quite common.
>
> > l.74 "Excessive invariance": in a classification setting, I believe it will almost always be the case that points outside the data distribution will be classified with high confidence, simply by virtue of the fact that there is a full-dimensional region of inputs for each label, whereas the data is typically lower dimensional.
>
> We believe that there is a misunderstanding. Importantly, we borrowed the term "excessive invariance" from [13], where it refers to misclassifying **in-distribution** examples with high confidence not those outside the training distribution. With the perspective of gradient starvation, we conjecture that "excessive invariance" can be attributed to the over-estimation of some dominant features while under-estimating others.
>
> > Experimental results: does "vanilla cross-entropy" include the weight-decay regularization?
>
> No, it does not include weight-decay. We will include experimental results with weight-decay showing that weight-decay does not improve the performance in any of the experiments. We will clarify this point in the text.
>
> > l.235: "Correlated with .9 probability"? What does this mean?
>
> It means that in 90 % of the training examples, color is correlated with the label: P(label = 0| color = green) = P(label = 1| color = red) = 0.9.
> This probability is 0.1 for the test set which equivalently means that the color is **inversely** correlated with the label in 90 % of the test examples: P(label = 0| color = red) = P(label = 1| color = green) = 0.9.
>
> > In Table 2, why is REx not discussed?
>
> We thank the reviewer for pointing out this shortcoming. We mistakenly omitted the discussion on REx during editing but will include it in the final version. In short, REx is a training method that encourages learning “invariant” representations. It is very similar to IRM and requires access to multiple training environments to quantify the concept of “invariance”. While REx scores are close to the ones obtained with our spectral decoupling (SD), we point out that SD does not require explicitly distinct training environments and thus is a more flexible method.
>
> ---
>
> Once again, we appreciate the time you took to provide us with such a detailed review. Please let us know if any further clarification is needed.

---

> > ### Author Response · Authors · 2021-08-19
> > **Follow up**
> >
> > Dear reviewer,
> >
> > We would like to thank you once again for your time and reiterate that we are keen to hear your thoughts after reading our responses. We will, of course, remain at your disposal should you have any further questions and continue to provide clarifications as promptly as possible.

---

### Author Response · Authors · 2021-08-11
**To all the reviewers**

We thank the reviewers for their insightful remarks that help to improve our manuscript considerably. Below, we first focus on the comments raised by most of the reviewers and then address each reviewer individually.

----

### On Primal (parameter space) vs. Dual (feature space) dynamics:
Although the cross-entropy loss is convex, it does not admit an analytical solution, even in a simple logistic regression [2]. Importantly, it also does **not** have a finite solution when the data is linearly separable [3] (which is the case in high dimensions [4]). As such, our study is concerned with characterizing **the solutions** that the training algorithm converges to. A dual optimization approach enables us to describe these solutions in terms of contributions of the training examples [5]. While primal and dual dynamics are not guaranteed to match, the solution they converge to is guaranteed to match [6], and that is what our theory builds upon.

For further intuition, we have already provided a simple experiment in appendix B.1, directly visualizing the primal vs. the dual dynamics as well as the effect of the proposed spectral decoupling method. As requested by most of the reviewers, we make sure to bring it up to the text.

----

### The intuition behind Spectral Decoupling (SD):
Consider a training datapoint x in the middle of the training process. Intuitively, the model has two options for decreasing the loss of this example:
1) Get more confident on a feature that has been learned already by other examples. or,
2) Learn a new feature.

SD, a simple L2 penalty on the output of the work, would favor (2) over (1). The reason is that (2) does not make the network over-confident on previously learned examples, while (1) results in over-confident predictions. Hence, SD encourages learning more features by penalizing confidence. Our principal novel contribution is to characterize this process formally and to theoretically and empirically demonstrate its effectiveness.

----

### On Squared-error loss vs. Cross-entropy:
#### Does Gradient Starvation happen for Squared-error loss?
As discussed in detail (see current appendix), we identify basic differences between squared error loss and cross-entropy. We compare the dual form of the squared-error, hinge, and cross-entropy loss in Tab. 5.  We note that for the squared-error loss, $\alpha_i$'s do not mix and hence are not coupled. That induces that no gradient starvation happens for squared-error loss. [7] refer to this as "independent mode learning".

#### Then why not use Squared-error loss for classification too?
The biggest obstacle when using squared-error loss for classification is how to select the target. For example, in a cats vs. dogs classification task, not all cats have the same amount of "catty features". However, recent results favor using squared-error loss for classification and show that models trained with squared-error loss are more robust [8]. We conjecture that the improved robustness can be attributed to a lack of gradient starvation.

----

### On using NTK:
Theoretical analysis of neural networks in their general form is challenging and generally intractable. Neural Tangent Kernel (NTK) has been an important milestone that has simplified theoretical analysis significantly and provides some mechanistic explanations that are applicable in practice. Inevitably, it imposes a set of restrictions; mainly, NTK is only accurate in the limit of large width. Therefore, the common practice is to provide the theoretical analysis in simplified settings and validate the results empirically in more general cases (see, e.g. [9, 10, 11]).
In this work, we build on the same established practices: Our theories analytically study an NTK linearized network; and we further validate our findings on several standard neural networks. In fact, in **all** of our experiments, learning is done in the regular "rich" (non-NTK) regime, and we verify that our proposed method, as identified analytically, mitigates learning limitations.

----

###  More context on this work:
This work has benefited from previous review cycles, which helped make our results more complete. We note that several results and discussion points are currently present in the appendix, many of which directly address reviewer questions. We give explicit details in our review replies and indicate where we intend to modify the main text. Finding a balance between main and supplemental arguments is a challenge with conference cycle review processes because there is little review continuity and fluctuating reviewer preferences. We kindly ask reviewers to take this into account when assessing the maturity and quality of our work.

----

###  References throughout this rebuttal:
[1] [Kernel Methods by Tengyu Ma and Andrew Ng](http://cs229.stanford.edu/notes2020fall/notes2020fall/cs229-notes3.pdf)\
[2] [Logistic Regression and Boosting for Labeled Bags of Instances by Xin Xu and Eibe Frank](http://citeseerx.ist.psu.edu/viewdoc/download?doi=10.1.1.148.993&rep=rep1&type=pdf)\
[3] [On the existence of maximum likelihood estimates in logistic regression models by  A. Albert and J. A. Anderson](http://citeseerx.ist.psu.edu/viewdoc/download?doi=10.1.1.470.9823&rep=rep1&type=pdf)\
[4] [Cover's theorem](https://en.wikipedia.org/wiki/Cover%27s_theorem)\
[5] [A dual coordinate descent method for large-scale linear SVM by Hsieh et al.](https://www.researchgate.net/profile/Kai-Wei-Chang-3/publication/312450680_A_dual_coordinate_descent_method_for_large-scale_linear_SVM/links/5ee0931a92851cf1386f5834/A-dual-coordinate-descent-method-for-large-scale-linear-SVM.pdf)\
[6] [Uniqueness of the SVM Solution by Christopher J.C. Burges and David J. Crisp](http://citeseerx.ist.psu.edu/viewdoc/download?doi=10.1.1.42.1546&rep=rep1&type=pdf)\
[7] [Exact solutions to the nonlinear dynamics of learning in deep linear neural networks by Saxe et al.](https://arxiv.org/abs/1312.6120)\
[8] [Evaluation of Neural Architectures Trained with Square Loss vs Cross-Entropy in Classification Tasks by Like Hui and Mikhail Belkin](https://arxiv.org/abs/2006.07322)\
[9] [Why Do Deep Residual Networks Generalize Better than Deep Feedforward Networks? — A Neural Tangent Kernel Perspective by Huang et al.](https://proceedings.neurips.cc/paper/2020/file/1c336b8080f82bcc2cd2499b4c57261d-Paper.pdf)\
[10] [A Generalized Neural Tangent Kernel Analysis for Two-layer Neural Networks by Chen et al.](https://arxiv.org/pdf/2002.04026.pdf)\
[11] [When and why PINNs fail to train: A neural tangent kernel perspective by Wang et al.](https://arxiv.org/pdf/2007.14527.pdf)\
[12] [Probabilistic Kernel Regression Models by Tommi S. Jaakkola, David Haussler](http://proceedings.mlr.press/r2/jaakkola99a.html)\
[13] [Excessive Invariance Causes Adversarial Vulnerability by Jacobsen et al.](https://arxiv.org/abs/1811.00401)\
[14] [Wide Neural Networks of Any Depth Evolve as Linear Models Under Gradient Descent by Lee et al](https://arxiv.org/abs/1902.06720)\
[15] [A Convergence Theory for Deep Learning via Over-Parameterization by Zhu et al.](https://arxiv.org/pdf/1811.03962.pdf)\
[16] [Removing Bias in Multi-modal Classifiers: Regularization by Maximizing Functional Entropies by Gat et al.](https://arxiv.org/pdf/2010.10802.pdf])\
[17] [In Search of Lost Domain Generalization by Ishaan Gulrajani and David Lopez-Paz](https://arxiv.org/pdf/2007.01434.pdf)\
[18] [Distributionally Robust Neural Networks for Group Shifts by Sagawa et al.](https://arxiv.org/pdf/1911.08731.pdf)\
[19] [Deep learning theory lecture notes by Matus Telgarsky ](https://mjt.cs.illinois.edu/dlt/index.pdf)\
[20] [Stability_theory](https://en.wikipedia.org/wiki/Stability_theory)\
[21] [The Implicit Bias of Gradient Descent on Separable Data by Soudry et al.](https://arxiv.org/abs/1710.10345)\
[22] [Machine learning for Covid-19 diagnosis; University of Cambridge](https://healthcare-in-europe.com/en/news/machine-learning-for-covid-19-diagnosis-promising-but-still-too-flawed.html)\
[23] [Invariant Risk Minimization by Arjovsky et al.](https://arxiv.org/abs/1907.02893)

---

### Author Response · Authors · 2021-09-01
**Final note**

Dear reviewers,

Thanks to your constructive comments, we believe our paper is considerably improved, but we recognize that it is still on the boundary of acceptance. There have been some misunderstandings that we have done our best to make clear. As we reach the end of the discussion period, we would like to encourage all the reviewers to refer to our responses. We appreciate it if you could acknowledge that you have read our replies. That would be reassuring for us to know that the final decision is made based on a transparent and clear understanding of our work.

We appreciate your time,\
The authors

---

### Decision · Program_Chairs · 2021-09-28

**Decision:**

Accept (Poster)

**Comment:**

The paper studies a phenomenon, referred to by as "gradient starvation" (e.g., https://arxiv.org/abs/1809.06848), in which only a subset of features relevant for the task is captured during the training, despite the presence of other predictive features. The reviewers found the explicit study of this phenomenon interesting and of practical relevance, and generally appreciated the numerical results provided in the paper. However, whereas some of the reviewers felt that the focus on the NTK regime is fair, others raised several significant concerns regarding the technical contributions of the work, in part (but not only) due to the by now well-established discrepancy between NTK and neural network models.

**Consistency Experiment:**

NeurIPS has a long history of experimentation. In 2014, NeurIPS ran an experiment in which 10% of submissions were reviewed by two independent committees to quantify the randomness in the review process. This year, we repeated a variant of this experiment to see how the quality of the review process has changed over time.  This paper was part of the experiment and was therefore assigned to two committees (consisting of reviewers, an Area Chair, and a Senior Area Chair) that reached independent decisions.  If both committees made the same recommendation, this recommendation was followed. If a single committee recommended acceptance, the paper was accepted (with the exception of a few cases in which the other committee identified what we considered a fatal flaw, e.g., an error in a key result).

This copy’s committee reached the following decision: **Reject**

The other committee assigned to the paper recommended **Accept (Poster)**.  You can find the other set of reviews, along with any follow up discussion with the authors here:
https://openreview.net/forum?id=h8flNv9x8v-